# Control of spatio-temporal variability of ocean nutrients in the East Australian Current

Megan Jeffers[1], Christopher C. Chapman[2], Bernadette M. Sloyan[2], and Helen Bostock[1]

[1]School of the Environment, Faculty of Science, The University of Queensland, Australia
[2]CSIRO Environment, Hobart Marine Laboratories, Hobart, Australia

**Correspondence:** Megan Jeffers (m.jeffers@uq.edu.au)

**Abstract.** The East Australian Current (EAC), the South Pacific's southward flowing western boundary current, dominates the marine environment of the east coast of Australia. Upwelling of deep EAC nutrient rich water into the oligotrophic surface waters is very important for maintaining upper ocean productivity. However, the role of EAC dynamics in upper ocean nutrient variability and resulting productivity is poorly understood. In this study we use physical and biogeochemical data collected from 2012–2022 to improve understanding of the variability of the nutrients in the upper water column at ∼27°S, a subtropical region strongly influenced by the EAC.

The 10-year data set shows that there is a seasonal increase in nutrient concentrations in the upper water column (0–200 m) in the austral spring (September–November) and autumn (March–May), and a minimum in winter (June–August). We also find that the nutrient concentrations in the upper water column are influenced by the position of the EAC jet. Two main modes of variability in the EAC's position are identified: an inshore mode with the jet flowing along the continental slope and; an offshore mode with the current core detached from the continental slope and flowing over the adjacent abyssal plain. The position of the EAC jet influences the location of upwelling of nutrient-rich water at depth (>200 m). For the EAC inshore mode, cooler, nutrient-rich waters are restricted to the area of the continental shelf and slope that is inshore of the EAC. The offshore mode exhibits a wider distribution of nutrient-rich waters over both the inshore shelf and slope and the offshore abyssal Tasman Sea. Our analysis highlights the important interactions between the highly variable EAC and the distribution of high nutrient waters, which has implications for primary production, fisheries, and the biological carbon pump.

## 1 Introduction

The East Australian Current (EAC) is the western boundary current (WBC) of the South Pacific subtropical gyre, and plays a significant role in the southwest Pacific Ocean circulation and nutrient distribution. The EAC transports warm water from the Coral Sea into the more temperate climate of the Tasman Sea (Archer et al., 2017a; Sloyan et al., 2016). The upper EAC waters (0–200 m) are oligotrophic (low nutrient concentration), however below 200 m the EAC transports nutrient rich water southwards (Suthers et al., 2011; Chiswell et al., 2015).

The EAC is fed by the South Equatorial Current via several westward flowing jets between 15°S and 23°S. The North Vanuatu and the North Caledonian jets flow into the northern Coral Sea at ∼15 °S and ∼19°S, respectively, providing inflowing

waters that are warm and salty, with low oxygen and nutrient concentrations (Kessler and Cravatte, 2013). At these latitudes the EAC has a relatively weak southward transport of between 6–8 Sv (Ganachaud et al., 2014; Kessler and Cravatte, 2013). Around 23°S, the addition of the sub-surface South Caledonian Jet strengthens the EAC flow to a mean southward transport of 24.6 Sv and also increases the subsurface oxygen concentrations (Kessler and Cravatte, 2013; Sloyan et al., 2016). This marks the start of the EAC jet "intensification zone" that extends southwards until the separation latitude at approximately 31–34°S (Oke et al., 2019; Ridgway and Dunn, 2003). At the separation latitude, the EAC bifurcates into two eddy-dominated flow paths, with one proceeding southward along the eastern Australian coast to Tasmania known as the south EAC extension, and the other flowing across the Tasman Sea to New Zealand as the Tasman Front or east EAC extension (Ganachaud et al., 2014; Godfrey et al., 1980; Oke et al., 2019; Ridgway and Dunn, 2003; Sutton and Bowen, 2014). The latitude of the separation point is significantly influenced by changes in the EAC further upstream (Cetina-Heredia et al., 2014; Li et al., 2021), emphasising the importance of understanding the EAC jet in this intensification zone.

The EAC exhibits seasonal changes in its physical flow. Several observational-based studies have revealed a seasonal change of approximately 9 Sv in the volume transport of the EAC, with the maximum occurring in austral summer and minimum in winter (Archer et al., 2017a; Godfrey et al., 1980; Ridgway and Godfrey, 1997; Sloyan et al., 2024). Temperature and salinity also show a seasonal cycle when considering the surface waters (potential density $\rho_0 >26.2$ kg/m$^3$), which are warmer and fresher during the summer (Oke et al., 2019; Sloyan et al., 2016). Additionally, at 29°S, Everett et al. (2014) identified a seasonal cycle in dissolved nutrients with increased dissolved nitrate and silicate concentrations in austral winter and spring (June - September).

Between the latitudes of 25–30°S the EAC, whilst maintaining a more coherent jet-like flow, is highly dynamic due to interactions with mesoscale eddies (Archer et al., 2017a; Oke et al., 2019) and larger scale forcing (Sloyan and O'Kane, 2015; Bull et al., 2020). This meandering of the jet can cause significant changes to the vertical structure of the current and southward transport (Roughan et al., 2022b; Sloyan et al., 2016). Previous work at ∼27°S identified a meander in the EAC jet which shifted the position of the main poleward (southward) flow laterally, on and off the continental slope (Sloyan et al., 2016). Similar variability and jet meandering has been observed in other WBCs. The Kuroshio Current experiences variability in its core position and strength, and experiences meandering and interactions with bathymetry and mesoscale eddies (Ebuchi and Hanawa, 2003; Kawabe, 2005, 1995; Waseda et al., 2003). The Florida Current is the upstream portion of the Gulf Stream, and has a very similar jet structure to the EAC (Archer et al., 2018), and exhibits a similar meandering behaviour, with meanders occurring on a time-scale of 3–30 days (Archer et al., 2017b). Other WBCs also experience meandering, including the Gulf Stream (Mao et al., 2023), Agulhas Current (Goschen et al., 2015), and the Brazil Current (Da Silveira et al., 2008).

Such current variability has been linked to biogeochemical variability in the EAC and other WBC systems. For example, in the EAC, Chapman et al. (2024) showed observations of nutrient injection into the surface layers during meandering, and that the vertical velocities that drove this were catalysed by the interaction of EAC with mesoscale eddies. Similarly, in the Kuroshio, the large meander mode results in an uplift of the nutricline, which increases nutrient availability in the near surface waters (Hayashida et al., 2023). In the upstream region of the Kuroshio, increased speeds of the current causes an uplift of nutrients onshore (Chen et al., 2022). Additionally, during periods where the Kuroshio sits closer to the coast, it interacts with

bathymetry which causes strong vertical mixing and uplift of nutrients to the continental shelf (Durán Gómez and Nagai, 2022). Mesoscale variability in the Florida Current is linked to the upwelling of cool, nutrient rich waters between the shelf break and the offshore meander (Fiechter and Mooers, 2007; Kourafalou and Kang, 2012). Upwelling has also been linked to the movement of jet meanders in the Agulhas Current, with upwelling occurring when the current shifts onto the continental shelf, or shifts offshore (Goschen et al., 2015). It is clear that meandering of WBCs upstream of their separation points is not unique to the EAC, and the current meandering can result in upwelling, which has been observed in several WBC systems.

Upwelling of deeper EAC nutrient rich water into the oligotrophic surface waters influences the primary production patterns, plankton community composition, and nutrient utilisation strategies of marine organisms along the east coast of Australia (Everett et al., 2014; Hassler et al., 2011). Previous work has shown that the EAC separation point is characterised by upwelling, resulting in an enhanced nutrient supply and increased biological activity, contributing to more productive fisheries (Everett et al., 2011; Figueira and Booth, 2010; Hassler et al., 2011; Olson, 2001; Suthers et al., 2023). The upwelling of nutrients to the surface layer in this region has been attributed to various factors, including wind forcing, EAC divergence, and the presence of eddies (Everett et al., 2011; Gibbs et al., 1998; Godfrey et al., 1980; Rochford, 1975; Roughan and Middleton, 2002, 2004). Some studies attribute the increased nutrient supply to the higher velocity of the current induced by topographic forcing (Blackburn and Cresswell, 1993; Boland and Church, 1981; Oke and Middleton, 2000), while other studies attributed the increased nutrient concentrations to the action of the EAC separating from the coast (Tranter et al., 1986). At the separation point, it has been suggested that shifts in the position of the EAC are associated with periods of increased nutrient upwelling (Rochford, 1975), resulting in the transport of nutrient-rich waters onto the continental shelf and coastal areas (Everett et al., 2014; Oke and Middleton, 2001; Roughan and Middleton, 2004). Roughan and Middleton (2002) find that when the EAC shifted towards the coast, nutrient upwelling to the surface occurred alongside uplifted isotherms. This upwelling supports considerable productivity in coastal and shelf regions in an otherwise relatively low productivity region (Nieblas et al., 2009; Roughan et al., 2022a; Schaeffer et al., 2013). While there is evidence that EAC dynamics influence nutrient supply to the euphotic upper ocean (Chapman et al., 2024), our understanding of nutrient dynamics within the EAC is limited, largely due to a lack of data (Everett et al., 2011, 2014; Hassler et al., 2011; McGillicuddy Jr, 2016; Oke and Middleton, 2001; Rocha et al., 2019; Schaeffer et al., 2016).

In this study we examine the spatial and temporal variability of nutrients from the continental shelf to the offshore region at $\sim$27 °S. We use 10-years of nutrient bottle data collected between 2012–2022 (Sloyan et al., 2016, 2024). We examine the seasonality of the nutrients and the role of the position of the EAC, relative to the continental shelf and slope, in influencing the distribution of nutrients in the upper water column. Our analysis highlights the important interactions between nutrient concentrations and distribution and the highly variable EAC, which has implications for primary production, fisheries, and the biological carbon pump. Understanding the dynamical implications of the position of EAC on nutrient distribution is essential for elucidating the broader implications of the EAC's role on marine ecosystems and fisheries. This long-term dataset offers a valuable insight into the EAC's dynamical influence on the surface and mixed layer oceanography, biogeochemical cycling, and nutrient concentrations.

## 2  Methods

### 2.1  Data

#### 2.1.1  The EAC Moorings

To capture the behaviour of the EAC in the intensification zone where it has a defined jet-like structure, a "picket-fence" mooring array was established by CSIRO and the Australian Integrated Marine Observing System (IMOS) at approximately 27°S, offshore of Brisbane, Australia (Sloyan et al., 2024). The mooring array was operational from 2012 to 2022, except for a 22-month period between 2013 and 2015. The EAC mooring array extends from 153.5°E to 155.5°E, covering a depth range of approximately 60 m to 5000 m (Figure 1; Sloyan et al. (2024)). The moorings were equipped with acoustic Doppler current profiling (ADCP) instruments that provided a vertical profile of horizontal current velocity and discrete temperature and salinity instruments at various vertical intervals. The data used in this study is the daily mooring velocity data product with missing or bad velocities "filled" using a machine learning approach (Sloyan et al. (2023, 2024); https://doi.org/10.25919/10h0-yf37).

#### 2.1.2  Conductivity, Temperature, Depth and Hydrochemical Data

For the assessment of the EAC's physical and biogeochemical properties, we utilise 162 Conductivity, Temperature, Depth (CTD) and Niskin bottle profiles collected during research voyages (Figure 1, Table 1). Of the 162 CTD stations, nutrient samples were collected on 136. CTD profiles provide measurements of temperature, salinity, and dissolved oxygen at 1 dbar pressure intervals. Water samples for nutrient analsyes were collected by Niskin bottles at discrete depths that span the entire water column. These water samples were analysed for nitrate, phosphate, and silicate (and nitrite and ammonia that are not used here). No voyages were conducted during austral summer (November–March) in the region to avoid cyclone season. However all months from April to November were sampled, providing observations for austral autumn, winter, and spring seasons over the 10-year period. All CTD data is available from the CSIRO Data Trawler (https://www.cmar.csiro.au/data/trawler/).

Although the CTD data are full depth (depths >4500 m over the abyssal plain), we use data from the upper 200 m of the water column, as this depth range spans the EAC jet core (Sloyan et al., 2024). This depth range also includes significant phytoplankton production, which is at a maximum in the Coral and Tasman Sea at a range of depths between 40–100 m (Ellwood et al., 2013). In the Southwest Pacific plankton tend to be nitrate and phosphate limited (Ellwood et al., 2013; Ustick et al., 2021; Hassler et al., 2011; Doblin et al., 2016). The Southwest Pacific contains a low dissolved phosphorus region centred around 28°S (Martiny et al., 2019). However, $NO_3$:$PO_4$ ratios show that nitrate is still the primary limiting nutrient (Hassler et al., 2011), particularly in the top 200 m (Doblin et al., 2016). Silicate is also a key nutrient in this region, as siliceous diatoms dominate the phytoplankton community (Eriksen et al., 2019; Thompson et al., 2009). However, like nitrate and phosphate, silicate is also limited in this region, and is experiencing a decline (Ellwood et al., 2013; Thompson et al., 2009).

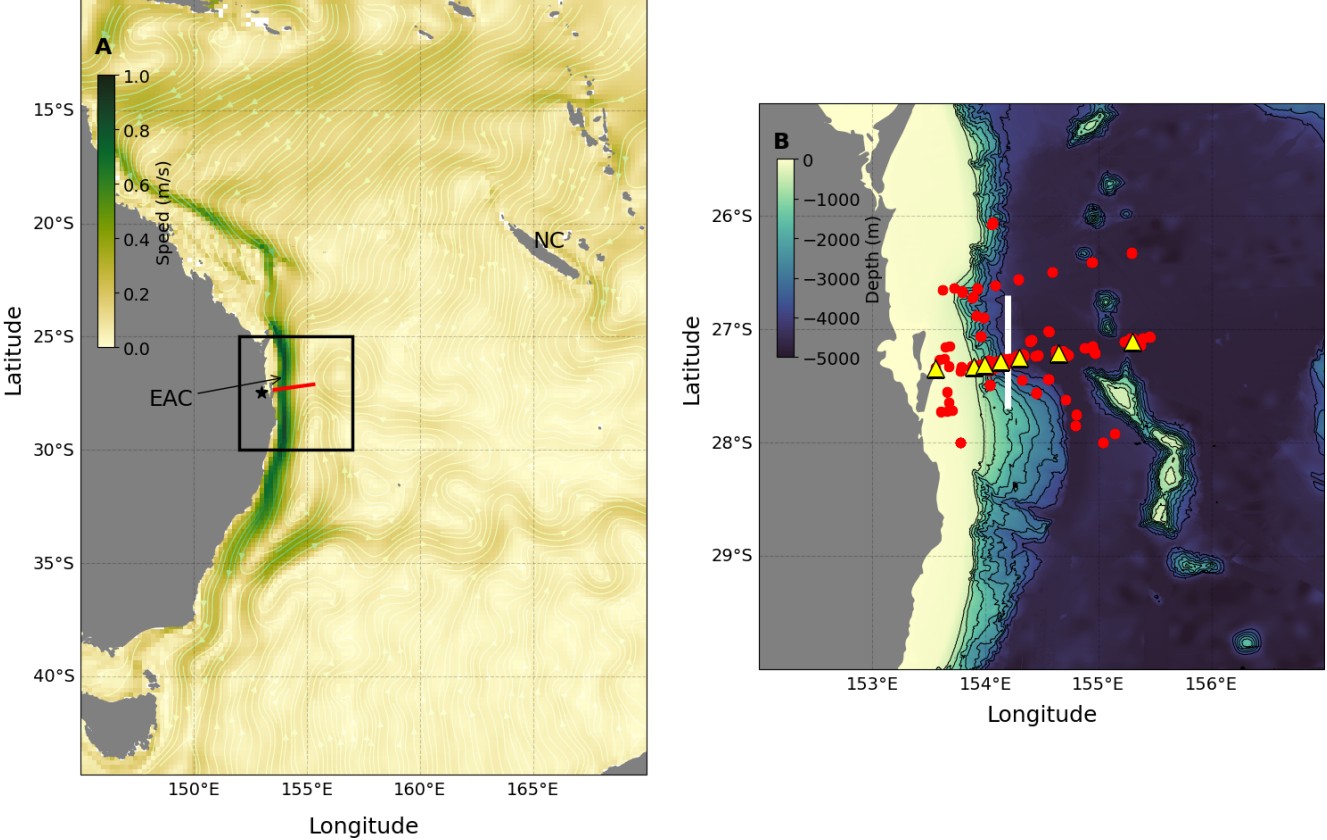

**Figure 1.** Location of data in context of broader surroundings. Panel (a) shows 20-year time mean (Jan 2000 – Dec 2019) of surface geostrophic currents from IMOS OceanCurrent. The location of the EAC mooring array is indicated by the red line. The black star shows the location of Brisbane, Australia and NC=New Caledonia. The black square outlines the region panel (b). Panel (b) shows the location of CTD stations (red dots) and EAC array moorings (yellow triangles) used in this study. The bold white line shows the position of the 154.2°E line used for defining EAC modes. The bathymetric map of the study area is also shown with the contour interval (black line) every 500 m from 500–4000 m water depth.

## 2.2 Analysis

We collate the 10-years of ship-based hydrographic data and analyse the physical (temperature and salinity) and biogeochemical (oxygen and nutrients) properties between 0–200 dbar across the longitudinal extent of the EAC mooring array. Mixed Layer depth (MLD) was calculated using density-based procedures developed by Holte and Talley (2009). This method uses an algorithm to choose the MLD from a suite of possible MLDs which are calculated using multiple threshold (difference from the surface) and gradient (where the depth-gradient exceeds a criteria) criteria. The threshold value of temperature and density for both the difference and gradient methods are $0.2°C$ and $0.03$ kg m$^{-3}$, and $0.005°C$ dbar$^{-1}$ and $0.0005$ kg m$^{-3}$ dbar$^{-1}$, respectively.

**Table 1.** List of voyages from which the CTD and nutrient data were used for this study. Including voyage ID, dates, research vessel and number of CTD and with hydrochemistry.

| Voyage ID | Voyage Dates | Vessel | No. CTD casts | No. CTD casts with hydrochemistry |
|---|---|---|---|---|
| SS2012_V01 | 20th April – 29th April 2012 | RV *Southern Surveyor* | 15 | 8 |
| SS2013_V05 | 20th August – 2nd September 2013 | RV *Southern Surveyor* | 25 | 25 |
| IN2015_V02 | 15th May – 26th May 2015 | RV *Investigator* | 17 | 16 |
| IN2016_V04 | 30th August – 23rd September 2015 | RV *Investigator* | 4 | 3 |
| IN2016_V06 | 28th October – 13th November 2016 | RV *Investigator* | 12 | 12 |
| IN2018_V03 | 29th April – 10th May 2018 | RV *Investigator* | 14 | 13 |
| IN2019_V05 | 9th September – 29th September 2019 | RV *Investigator* | 26 | 25 |
| IN2021_V03 | 7th May – 2nd June 2021 | RV *Investigator* | 28 | 16 |
| IN2022_V06 | 14th July – 28th July 2022 | RV *Investigator* | 21 | 18 |

The hydrographic data are grouped by the austral seasons of autumn (March–May), winter (June–August), and spring (September–November), and by mode position of the EAC jet (Figure 2). For each day that a CTD station was occupied, we classified the EAC into an "inshore mode" or an "offshore mode" based on the meridional velocity profiles and position of the EAC core from the mooring data. The EAC is considered to be in an inshore mode when the EAC core is located westward of 154.2°E (approximately over the continental slope, Figure 1 (b), bold white line), otherwise it is considered an offshore mode. Comparing the velocity structure of the "inshore" and "offshore" modes with Sloyan et al. (2016), we note that the inshore mode essentially corresponds to the climatological southward velocities, which we have confirmed by taking composite averages of the meridional velocity across all days where a CTD station was occupied and where time steps are classified as inshore mode (Figure 3 (b) and (e) respectively). For the time period considered here, 57% of samples are categorised as belonging to the inshore mode. In contrast, the "offshore mode" approximately matches mode 2 from Sloyan et al. (2016) and diverges from the climatological position of the EAC.

After grouping the hydrographic profiles by season and EAC jet position, we interpolate the temperature, salinity, oxygen and nutrient data to a 10 dbar vertical and 0.1° longitude grid (Figure 2) to produce property sections for each season (Figures 4 and 6) and EAC mode (Figures 8 and 9). To test for statistical significance a Monte Carlo simulation was run. For each variable, a random subset of data points were selected and used to make an interpolated pressure-longitude transect. This was repeated 1000 times and sections were constructed by selecting only the <5th and >95th percentile observations for each grid cell from the Monte Carlo. Stippling was added to all seasonal and mode sections for areas of statistical significance at the 95% confidence interval.

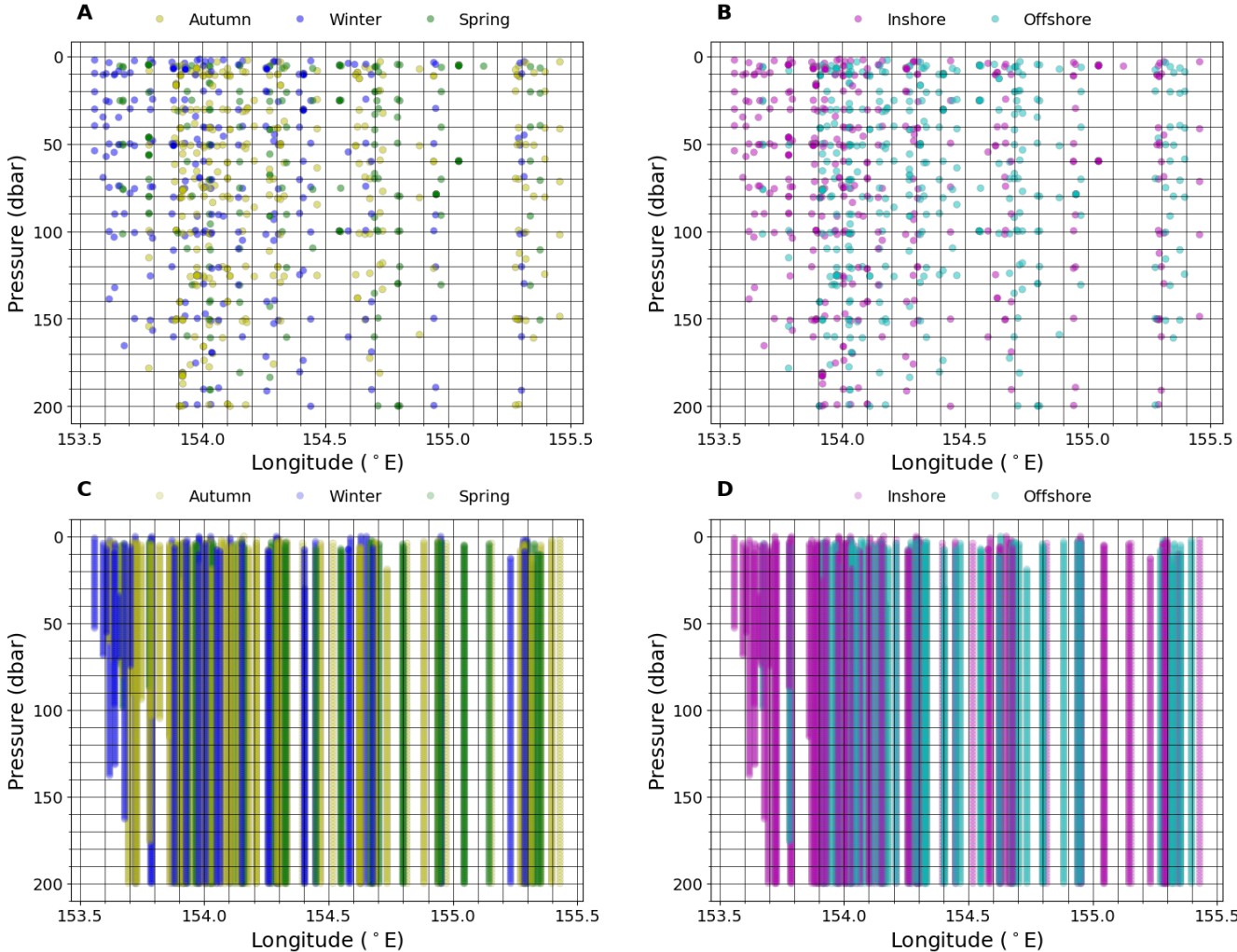

**Figure 2.** Full 10 years of hydrograhic data points (coloured dots and lines) and grid (black lines) used for interpolated property sections. Distribution of nutrient data (a) grouped by season (austral autumn (March–May), winter (June–August), and spring (September–November)), and (b) EAC mode, and distribution of CTD data (c) grouped by seasons, and (d) EAC mode.

## 3 Results

### 3.1 Seasonality

We will first focus on the seasonal variability of the physical and biogeochemical variables in the EAC during austral autumn (March–May), winter (June–August), and spring (September–November).

The upper 200 m of the water column shows in-situ temperature ranges between ∼15–25°C, practical salinity ranges between ∼35–35.8 and oxygen ranges between 160-230 $\mu$mol/L for all seasons (Figure 4). In general, water is warmest, freshest and

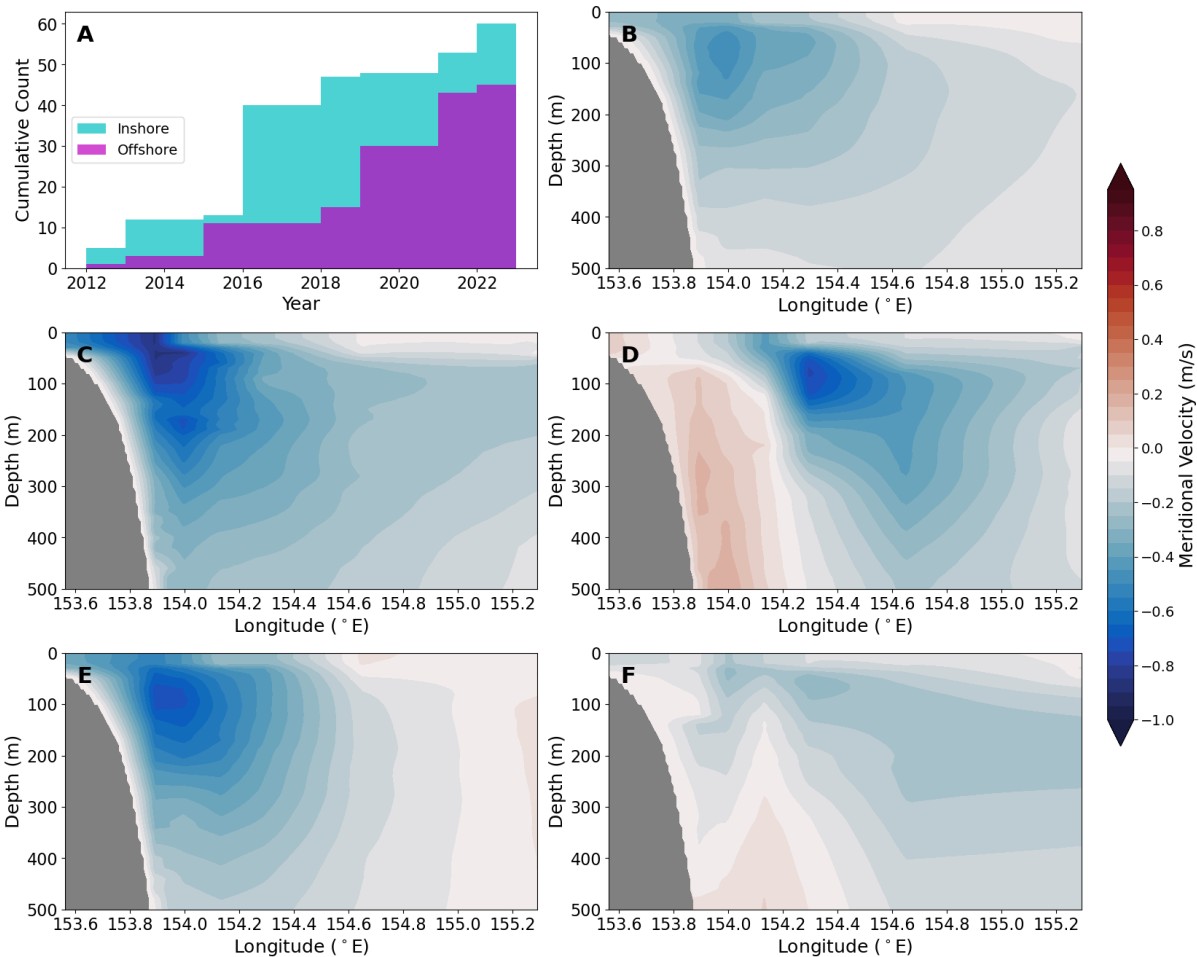

**Figure 3.** Panel (a) shows the cumulative sum of days observed for the inshore (blue) and offshore (purple) modes, considering only when a CTD station was occupied. Meridional velocity ($m/s$; positive is equatorward) plotted for the upper 500 m of the water column from the daily EAC mooring data product for (b) mean over all days with a CTD station. A one-day example for (c) inshore mode, and (d) offshore mode. Also plotted is mean EAC jet for all days classified as (e) inshore mode, and (f) offshore mode.

most oxygenated in the upper ~50 m and becomes cooler and reduced in oxygen with depth. The surface waters (0–100 dbar) are warmest in autumn, cooler during the winter period, and begin to rise slightly in temperature during spring. During both austral autumn and spring, there is a sharper gradient in both temperature, salinity, and oxygen between the surface and 200 dbar depth across the transect compared to winter. Oxygen concentrations above the mixed layer are lowest in autumn, increase in winter, and reach a maximum in spring. Below the mixed layer, the lowest oxygen concentrations are found near the western boundary and highest oxygen concentrations along the eastern edge of the section. Salinity is highly variable, with the near surface (above the mixed layer) having a minimum in autumn, increasing through winter, and reaching a maximum

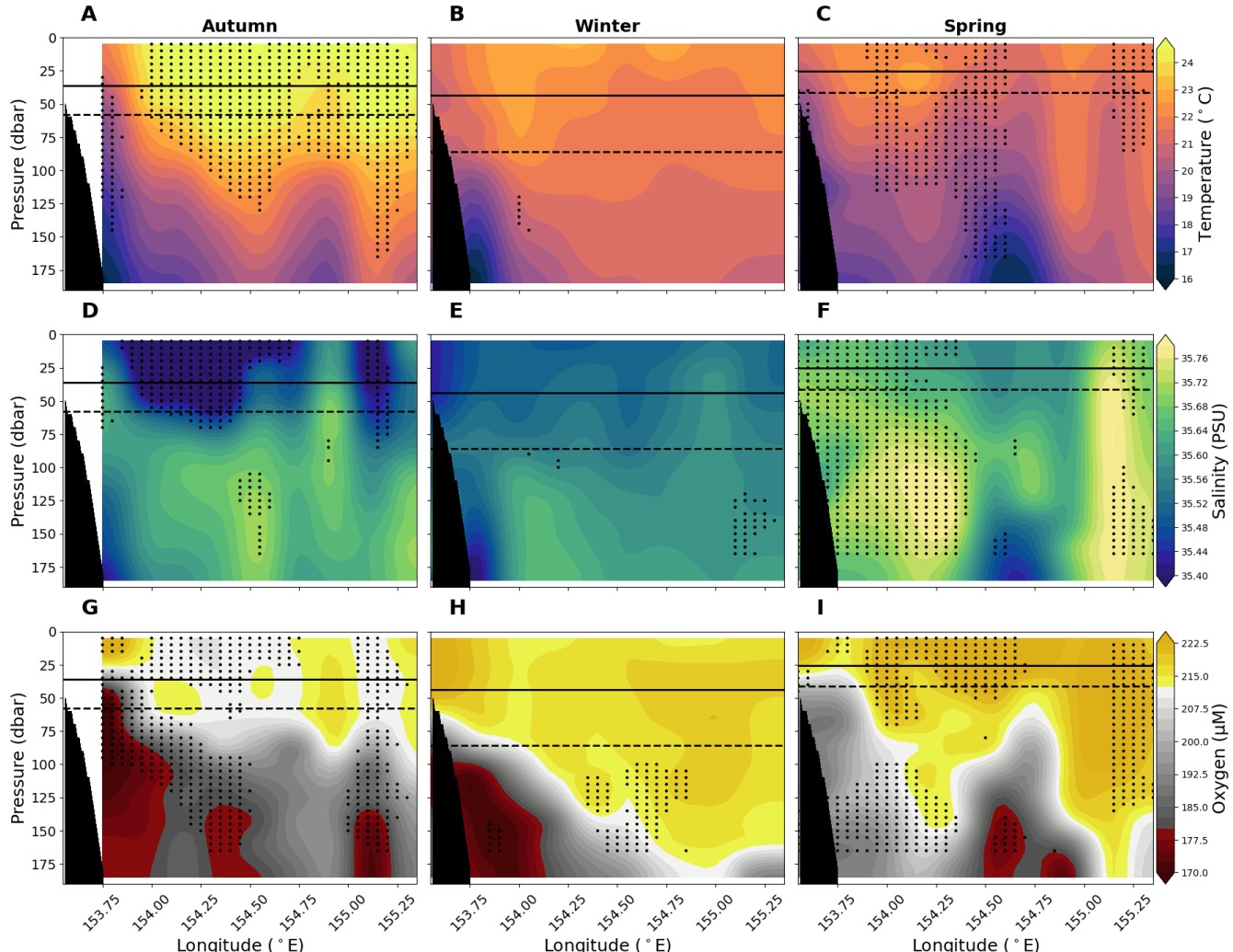

**Figure 4.** Vertical sections of (a), (b), (c) in-situ temperature (°C), (d), (e), (f) practical salinity, and (g), (h), (i) oxygen for the upper 200 dbar compiled from the 10 years (2012–2022) of data for austral autumn (March, April, May, left column), winter (June, July, August, middle column) and spring (September, October, November, right column). Solid lines show mean MLD and dashed lines show the 90[th] percentile MLD averaged across longitude. Stippling indicates statistical significance following the procedure described in the Analysis section (section 2.2). Black shading represents the seafloor.

in spring. The subsurface salinity (50–150 dbar) has highest values in spring. The winter seasonal changes in temperature, salinity and oxygen have little significant divergence from the mean state, however, the seasonal extremes of warm, fresh, low oxygen waters in autumn, and cool, saline, more oxygenated waters in spring are statistically significant, particularly in the near-surface.

165

There is a slight seasonal cycle in MLD, with a mean of 36.08 m in autumn (interquartile range (IQR) of 21.75–47.75 m, and 90th percentile 57.75 m), followed by a deepening in winter (mean MLD of 43.65 m, IQR of 16.50–62.75 m and 90th percentile of 85.80 m), and shoaling in spring (mean MLD of 25.40 m, IQR of 15.13–29.38 m and 90th percentile of 41.35 m).

The Temperature, Salinity, Oxygen (TS-O) diagrams show that the surface waters ($\rho_\theta < 26.4$) are warmer with slightly lower salinity in autumn, cooling and freshening in winter, and increasing in temperature and salinity again in spring (Figure 5). Surface waters are generally oxygen-rich, with oxygen decreasing in the subsurface waters. There is a slight increase in oxygen in the surface waters in winter and spring, likely due to slight cooling in the temperature. All seasons show anomalous low oxygen concentration within the surface layer, these are particularly evident in autumn (Figure 5).

Nutrient concentrations are generally low at the surface (0–50 dbar) and increase with depth (Figure 6). The ratio of nitrate to phosphate for 0–100 dbar and 100–200 dbar is 13.19 and 15.11, respectively. These values approximately agree with the Redfield ratio, particularly between 100–200 dbar. There is evidence of a subtle seasonal cycle in nutrients in the upper 200 dbar of the water column, as shown in Figure 7 (a)–(c). Within each season, there is a large degree of variability. In general, autumn and spring have similar IQR, and winter has the smallest IQR for all nutrients sampled. This suggests winter has the lowest nutrient concentrations available to the surface waters compared to the other seasons. Changes are broadly consistent across the three different nutrients, particularly in the variance and IQR. Although broadly similar, there are differences in the mean nutrient concentrations between the three nutrients with austral season. There is little change to the nitrate mean, although there is an increased number of observations with high concentration in autumn and spring compared to winter (Figure 7 (a)). Phosphate experiences a slight increase in the seasonal mean from autumn through to spring (Figure 7 (b)). Silicate has the largest mean in autumn and winter and the lowest mean in spring (Figure 7 (c)).

The nutrients also exhibit a seasonal change with the longitude. The western edge of the section (westward of approximately 154°E) has a similar distribution of nutrients throughout the water column during autumn and winter and a slight decrease in nutrient concentration during spring (Figure 6). The eastern section of the transect, particularly eastward of ~154.5°E, surface waters between 50–200 dbar are more nutrient-rich during autumn, with reduced nutrient concentration during winter, and show an increase in nutrient concentration again during spring, although with more variability (Figure 6).

## 3.2 Response of the ocean environment to EAC variability

Here, we present the physical and biogeochemical characteristics sorted by EAC inshore or offshore mode. The inshore mode (Figure 3 (c) and (e)) approximately follows the climatological position of the EAC, where meridional flow between 40–100 m is generally >0.4 $m/s$ in the poleward/southward direction and sits over the continental slope, with the majority of the current sitting westward of 154.2°E. The inshore mode has southward velocity over the continental slope and northward velocity further offshore at the eastern edge of the mooring array. In contrast, the EAC "offshore" mode (Figure 3 (d) and (f)) has an EAC jet that is displaced eastward from the continental slope and found to the east of ~154.3°E. In this "offshore" mode there is evidence for northward flow occurring inshore at depths below ~100 m. Both the inshore mode and the offshore mode occurred during all the seasons sampled, however, winter was dominated by the inshore mode, whereas autumn and spring

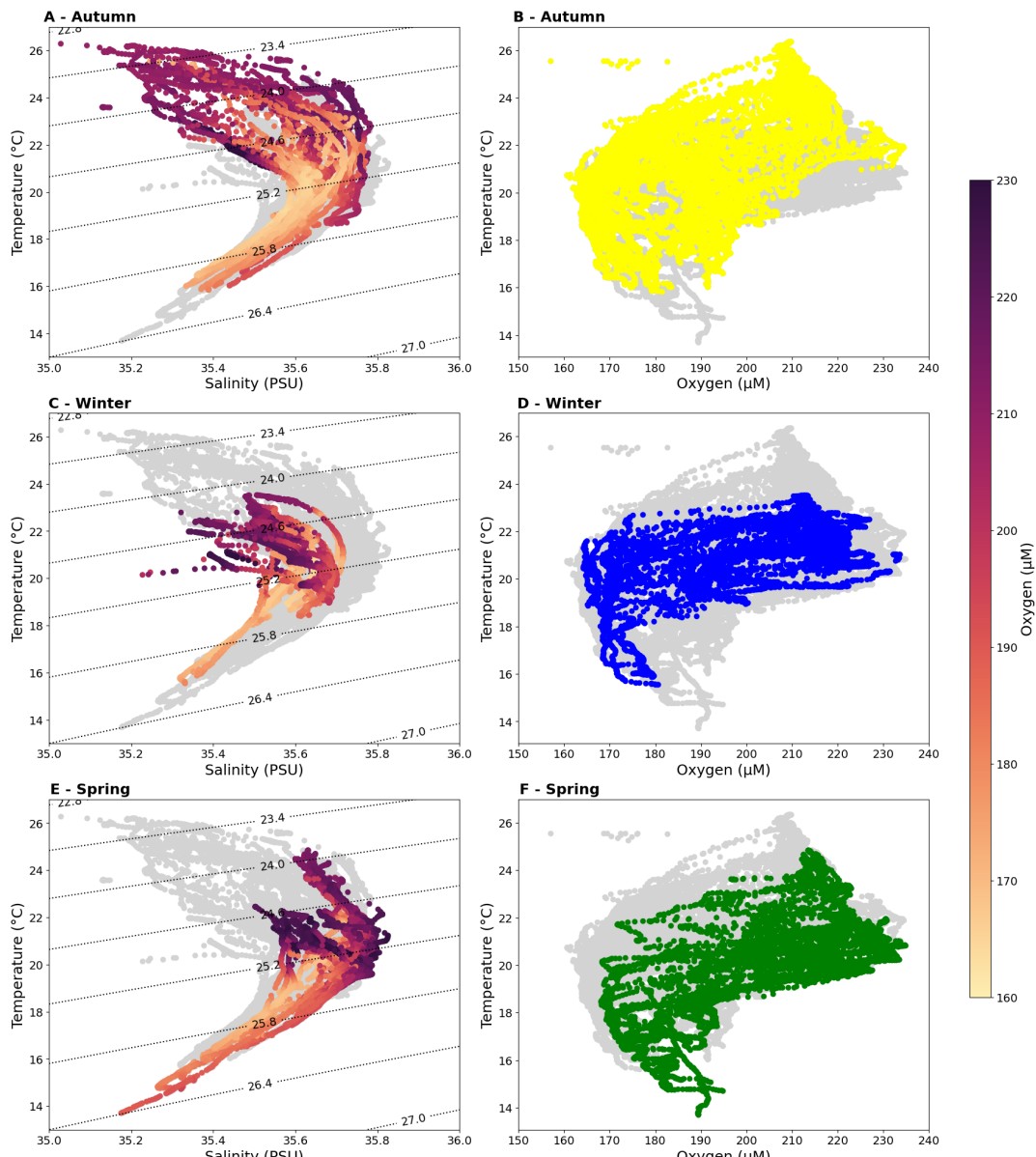

**Figure 5.** Temperature-Salinity-Oxygen (TS-O) diagrams for austral (a) autumn, (c) winter, and (e) spring with oxygen concentrations coloured. Temperature-Oxygen (T-O) diagrams for austral (b) autumn (yellow), (d) winter (blue) and (f) spring (green). Other CTD data are shown (light grey dots) in each figure, and potential density isopycnals (kg m$^{-3}$) are shown as black dotted lines on TS-O diagrams.

had a more even distribution of both modes (Table 2). Additionally, all voyages experienced a "switch" between the two EAC modes, except for the 2016 voyages, during which only the inshore mode was observed.

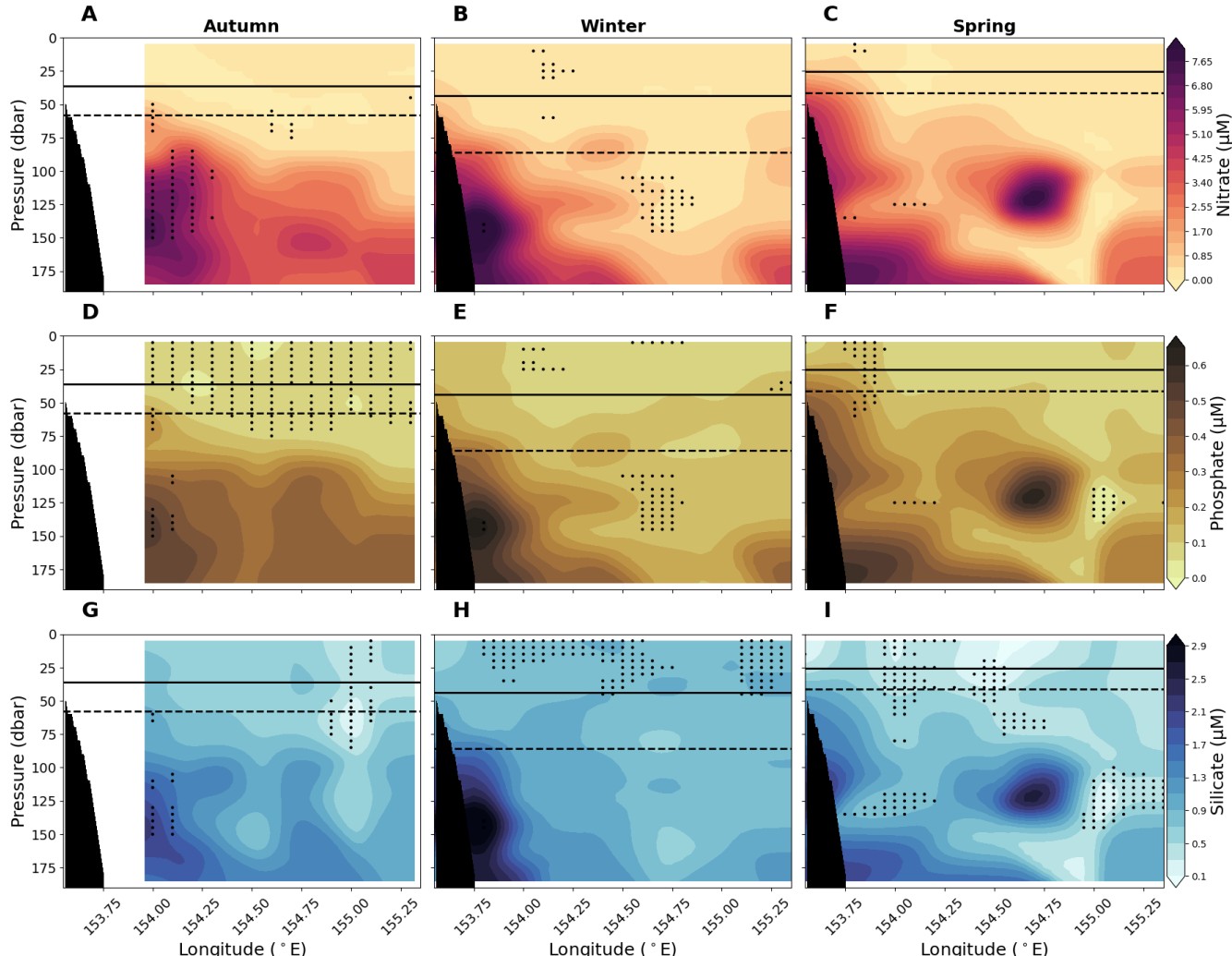

**Figure 6.** Vertical sections of (a), (b), (c) nitrate, (d), (e), (f) phosphate, and (g), (h), (i) silicate for the upper 200 dbar compiled from the 10-years (2012–2022) of data for austral autumn (March, April, May, left column), winter (June, July, August, middle column) and spring (September, October, November, right column). Solid lines show mean MLD and dashed lines show the 90[th] percentile MLD averaged across longitude. Stippling indicates statistical significance following the procedure described in the Analysis section (section 2.2). Black shading represents the seafloor.

Similarly to the seasonally grouped sections, EAC jet "inshore" or "offshore" mode sections have the warmest, freshest and most oxygenated water found in the upper ~50 dbar, and sections cool and have reduced in oxygen concentrations with depth (Figure 8). The TS-O diagram (not shown) reveals little discernible difference between the two modes, as the modes have similar temperature and oxygen ranges.

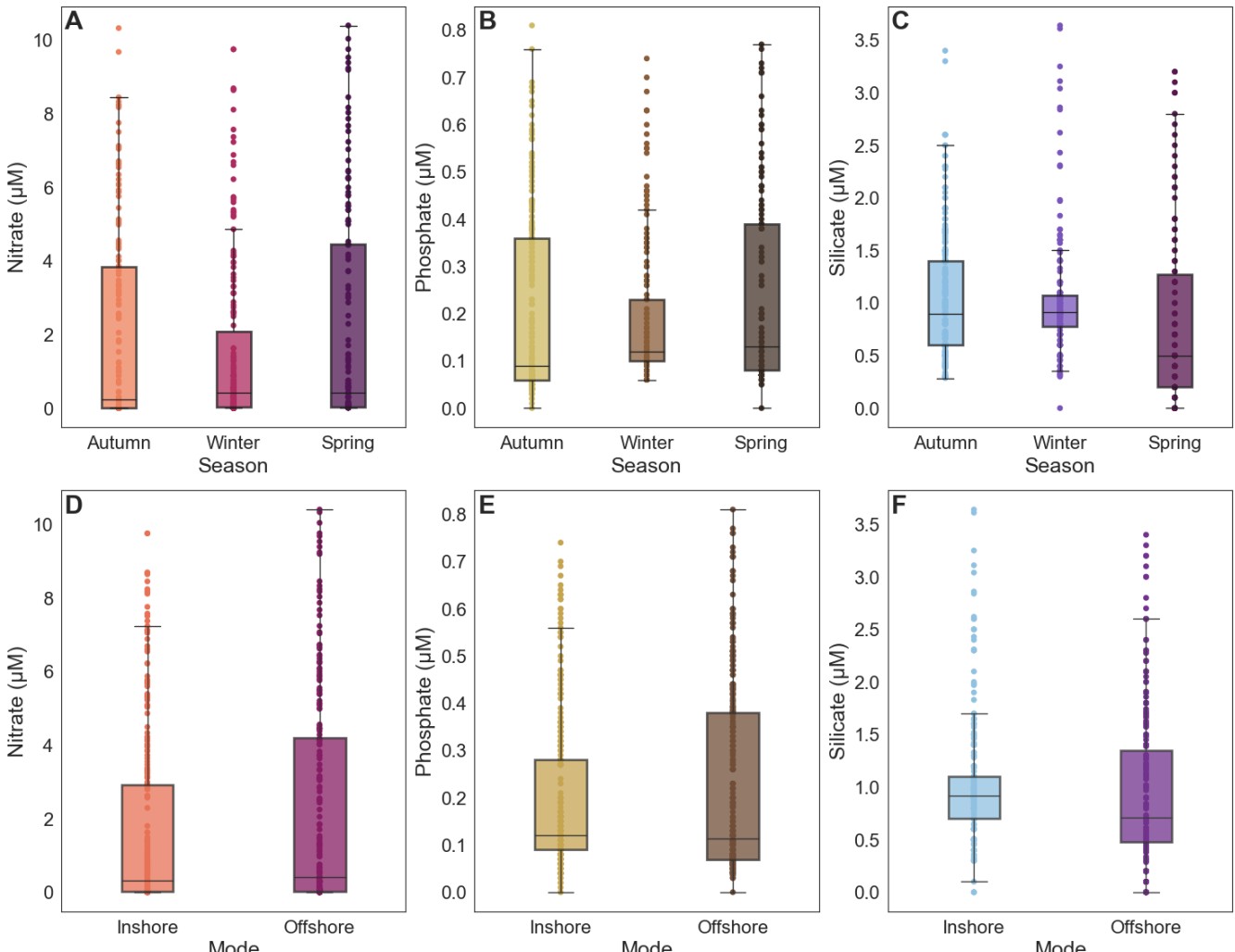

**Figure 7.** Box plots of nutrient concentrations across (a), (b), (c) the three seasons (Austral autumn, winter, and spring) and (d), (e), (f) the two modes sampled. Coloured dots are data points. Coloured boxes represent interquartile range (IQR), with the line inside the box representing the median ($50^{th}$ percentile). The whiskers (vertical black line) represents the range that contains 1.5 times the IQR above or below the first and third quartile.

As mentioned, the velocity profile of the inshore mode is similar to the climatological mean velocity profile of the EAC, thus, there are only a few significant changes to the properties of the water column. The offshore mode is cooler than the inshore mode across most of the section. However, for the western portion, over the continental slope and below the mixed layer, we find that the offshore mode is warmer, more saline, and more oxygenated than the corresponding locations of the inshore mode. These offshore mode property differences are generally significant for the portion of the transect that is west of

**Table 2.** Number of days sampled in each season and mode. The percentage of days categorised in each mode for every season is shown in brackets, as well as total percentage number of days sampled in each mode and season.

| Mode | Autumn | Winter | Spring | Total |
|---|---|---|---|---|
| Inshore | 16 (38%) | 17 (81%) | 27 (64%) | 60 (57%) |
| Offshore | 26 (62%) | 4 (19%) | 15 (36%) | 45 (43%) |
| Total | 42 (40%) | 21 (20%) | 42 (40%) | 105 |

the EAC core. There is no difference in mean MLD between the two modes (35.77 m vs 34.61 m), or in variability of MLD (IQR of 17.75–48.50 m vs 15.38–48.75 m). The 90th percentiles were 63.00 m vs 65.10 m.

When considering changes in nutrient concentrations for all data in the upper 200 dbar of the water column, there is no clear change in mean nutrient concentrations between the two modes, although the offshore mode has a larger IQR for all three
nutrients sampled (Figure 7 (d)–(f)). The larger IQR in the offshore mode results from all nutrients having greater variability, reflecting a larger amount of samples with high nutrient concentration. This suggests a greater availability of nutrients in the upper 200 dbar when compared with the inshore mode. Similar to the changes across the three seasons, changes across the two modes are similar between the three nutrients. Additionally, the differences in nutrient concentrations between the two modes are on the same order of magnitude as seasonal changes.

Differences in nutrient concentrations become clearer when considering the effect of depth and longitude (Figure 9). The nutrient sections based on EAC jet location show that generally nutrient poor water is found above the mixed layer during the inshore mode, and the highest nutrient concentrations occur at the western edge of the section at a pressure of approximately 150 dbar (Figure 9). For the offshore mode, waters are generally more nutrient-rich. Whilst there are not widespread statistically significant increases in nutrient concentrations between the two modes and the mean state, there is evidence for an increase in
the concentrations of nitrate and phosphate above and near the mixed layer during the offshore mode when compared to the inshore mode.

## 4  Discussion

In this study, we use physical and biogeochemical data collected over a 10-year period to determine an offshore seasonal cycle in nutrient concentrations, and reveal the effect of the varying position of the EAC on the availability of nutrients to the
230 surface waters in the upstream EAC system. The results suggest that there is a seasonal cycle in biogeochemical properties. Nitrate, phosphate, and silicate peak in concentration in autumn and spring below the mixed layer, but these are associated with different oxygen concentrations. There is a low-oxygen high nutrient peak in autumn and a higher-oxygen and nutrient peak in early spring. Previous observations from coastal sites have found a spring peak in nutrients at 29°S (Everett et al., 2014) and also at the North Stradbroke Island National Reference Site at 27°S (Butler et al., 2020). They also suggested an early winter

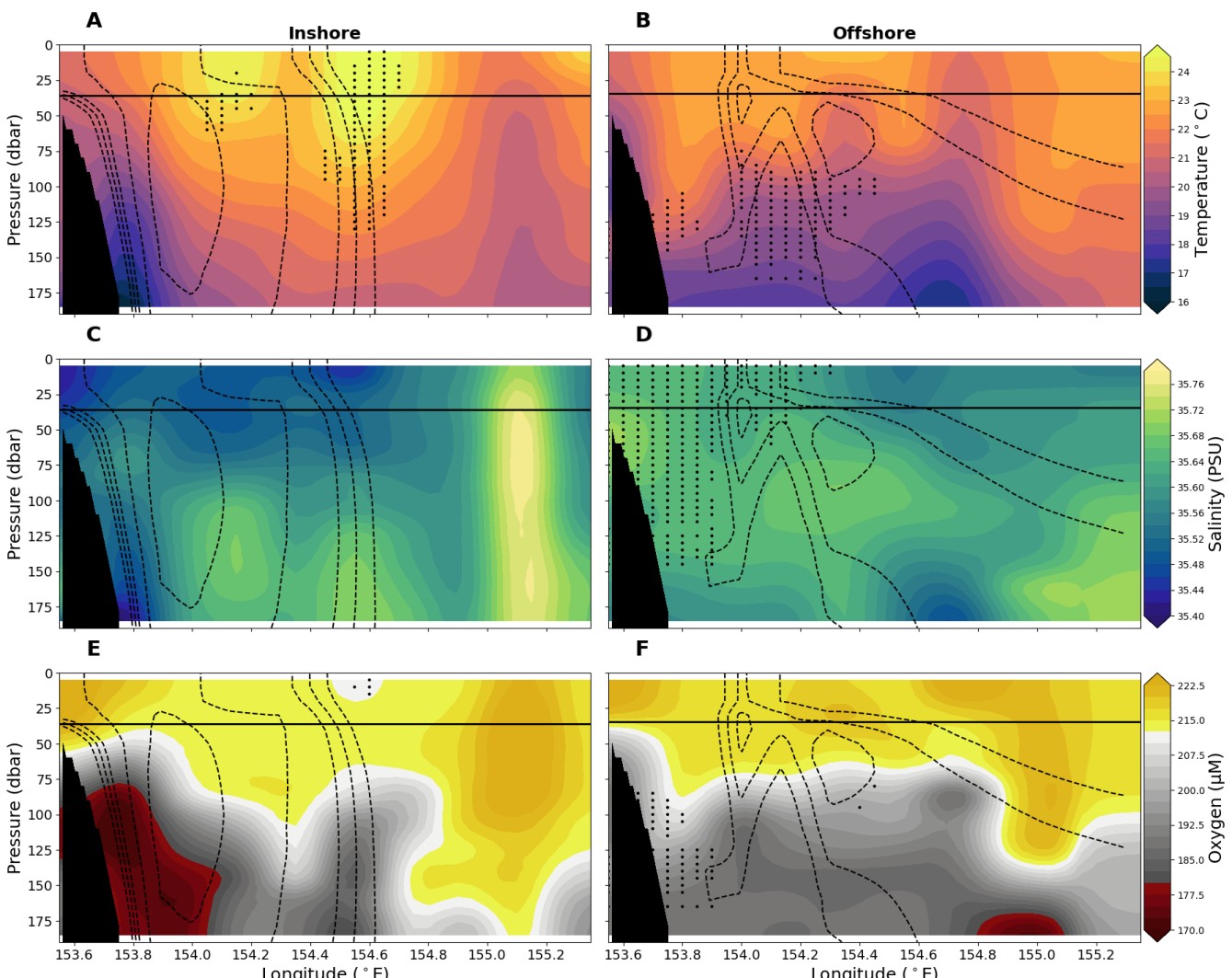

**Figure 8.** Vertical sections of (a), (b) in-situ temperature (°C), (c), (d) practical salinity, and (e), (f) oxygen for the upper 200 dbar compiled from the 10 years (2012–2022) of data for the two EAC jet modes — inshore mode (left column), and offshore mode (right column). Solid lines show mean MLD averaged across longitude. Dashed lines show the 0.6, 0.4, 0.25, 0.2, and 0.15 $m/s$ contours of mean southward velocity for the corresponding EAC mode. Stippling indicates statistical significance following the procedure described in the Analysis section (section 2.2). Black shading represents the seafloor.

minimum in silicate. Additionally, Rocha et al. (2019) found that the nutricline deepens by approximately 50 m during the summer, reducing availability of nutrients to the surface waters.

While we observe a seasonal nutrient signal, we also highlight the influence of the EAC core's position relative to the continental shelf/slope. The position of the EAC affects the vertical distribution of nutrients and their longitudinal gradients along

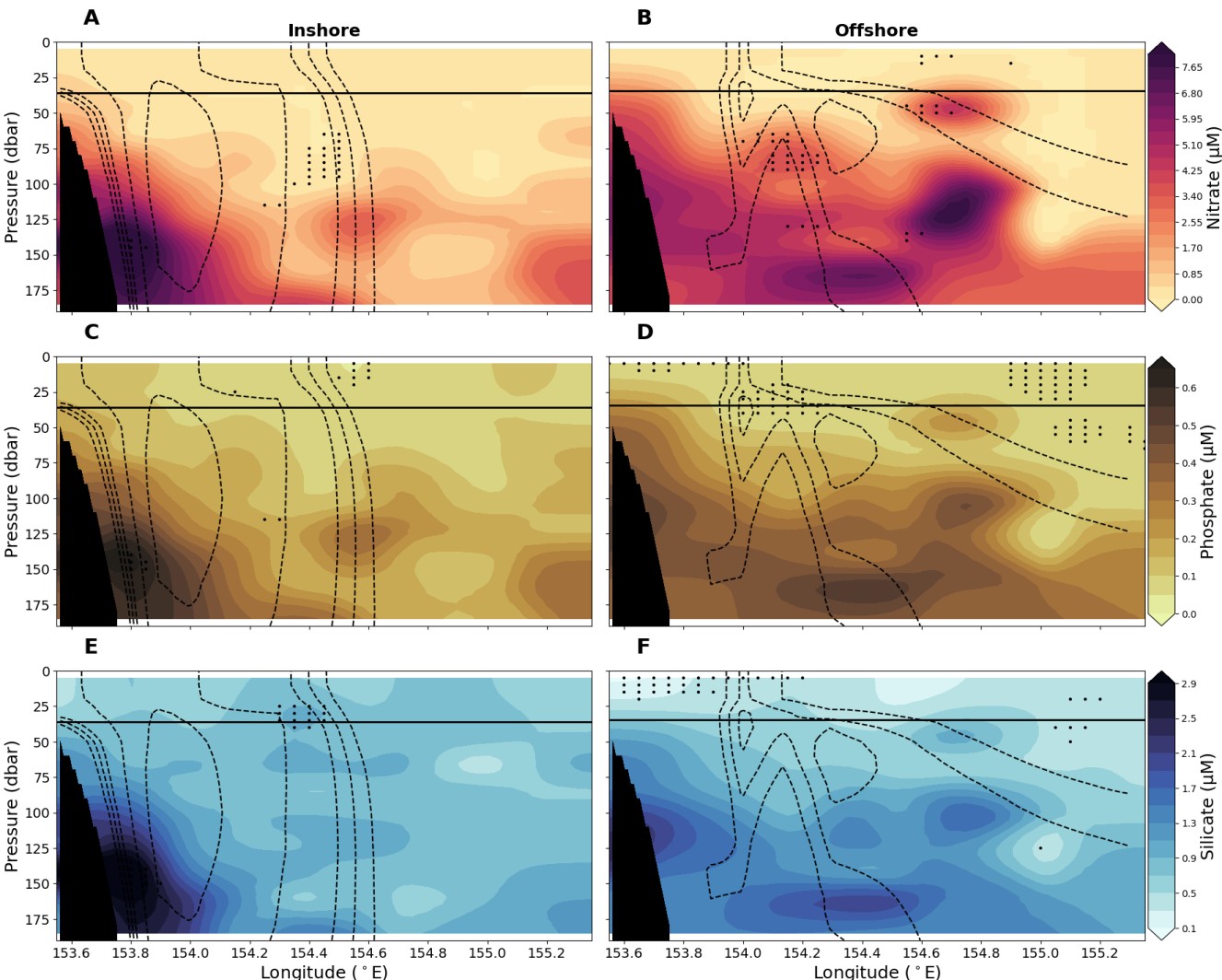

**Figure 9.** Vertical sections of (a), (b) nitrate, (c), (d) phosphate, and (e), (f) silicate for the upper 200 dbar compiled from the 10-years (2012–2022) of data for the two EAC jet modes — inshore mode (left column), and offshore mode (right column). Solid lines show mean MLD averaged across longitude. Dashed lines show the 0.6, 0.4, 0.25, 0.2, and 0.15 $m/s$ contours of mean southward velocity for the corresponding EAC mode. Stippling indicates statistical significance following the procedure described in the Analysis section (section 2.2). Black shading represents the seafloor.

the transect. For the inshore mode, cooler, fresher, oxygen poor, and nutrient-rich water occurs on the western (inshore) edge
of the property sections (west of 154°E), trapped between the EAC and continental slope. Whilst there is a statistical increase
in nutrient concentrations, due to the depth (150 dbar) and presence of the EAC jet, these nutrients would be inaccessible to
the surface layer. For the offshore mode, there is an uplift of nutrients across the transect, particularly offshore at ~154.7°E.

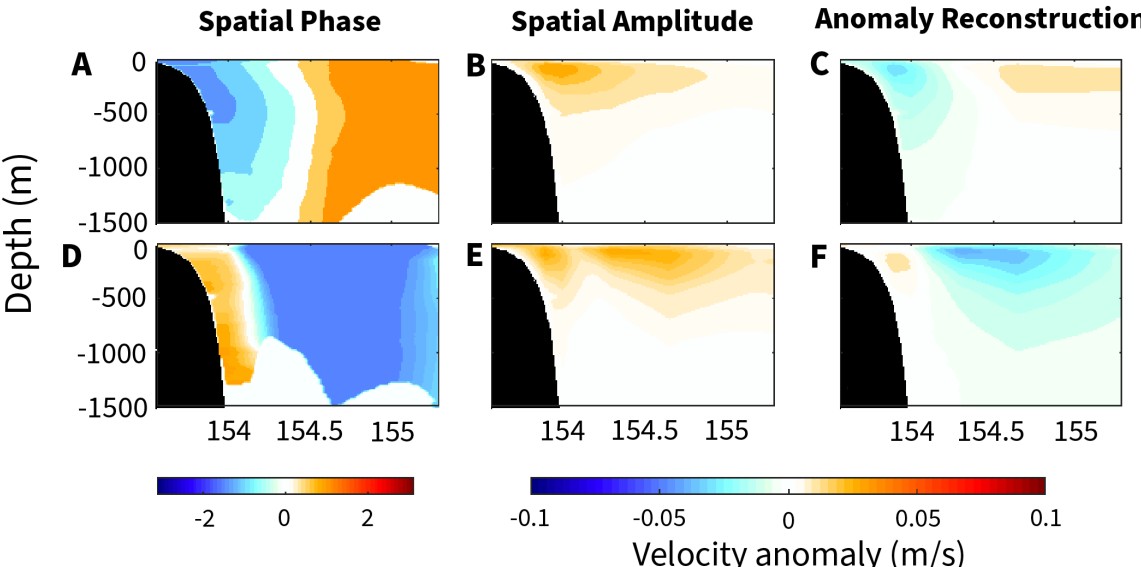

**Figure 10.** The EAC velocity anomaly of the first two modes of the Complex Empirical Orthogonal Function (CEOF) analysis. For the leading two modes, the spatial maps of (a), (d) phase $(-\pi - \pi)$ and (b), (e) amplitude and (c), (f) reconstructed velocity anomaly (m/s) are shown. The inshore mode approximately matches the CEOF mode 1 (a), (b), (c), which accounts for 65.4% of velocity anomaly. The CEOF mode 2 (d), (e), (f) fits the offshore mode, and accounts for 13.3% of velocity anomaly. The reconstructed velocity anomaly for each mode is determined from the spatial amplitude and phase maps.

Additionally, while the MLD is similar between the two modes, there is a shoaling of the nutricline and an increase in nutrients closer to and entering the mixed layer while the EAC is in the offshore mode compared to the inshore mode. The effect of the EAC core location highlights a potential mechanism for episodic nutrient supply to the surface layer. A complex empirical orthogonal function (CEOF) analysis of the southward velocity component shows that the onshore and offshore EAC modes explain greater than 78% of the velocity variance with periods of 120-60 days (Figure 10). The first two CEOF modes explain the wavering of the EAC across the mooring array, with the first mode showing the EAC located near the continental shelf, flowing over the abyssal plain, accounting for 65.4% of the velocity anomaly. The second mode shows an EAC detached from the continental shelf and accounts for 13.3% of the velocity anomaly. This highlights the variability in the EAC, separate from seasonality and large scale dynamics.

The distribution of nutrients that occur in the two EAC jet modes may be connected to different drivers. The encroachment of the EAC towards the continental shelf can cause upwelling along the slope (Schaeffer et al., 2013). Additionally, increasing EAC speeds has previously been linked to upwelling (Archer et al., 2017a; Oke and Middleton, 2001; Roughan and Middleton, 2002). In the Kuroshio, increased speeds where the current sits close to the shelf in its upstream region causes a similar uplift of nutrients onshore (Chen et al., 2022). As such, the inshore mode of the EAC – characterised by high speeds and a position

close to the continental shelf – may still experience some upwelling of nutrient rich waters, however this upwelling will be limited to the near slope/shelf area, trapped by the EAC position.

The drivers of the shift in the position of the EAC core, related to the offshore mode, are yet to be determined. Early studies implied seasonality plays a role in jet meandering (Ridgway and Godfrey, 1997), while more recent studies found seasonality plays no role in the core position (Archer et al., 2017a). It may be related to the energy of the EAC system, as a higher energy jet displays different dynamics to a low energy jet (Li et al., 2021). However, previous studies have not been able to attribute EAC wavering or meandering to one clear cause (Archer et al., 2017a; Bowen et al., 2005; Sloyan et al., 2016). Despite this, the effects of mesoscale eddy interactions are heavily implicated, as there are strong interactions between eddies and the mean flow path of the EAC (Li and Roughan, 2023). In the process of classifying EAC modes for this study, it was noted that the offshore mode is often associated with a cyclonic eddy sitting on the inshore side of the EAC, similar to what was observed by Chapman et al. (2024). As such, upwelling is likely linked to oceanic internal processes, such as eddy interactions and sub-mesoscale circulations (*e.g.*, Roughan et al., 2017; Roughan and Middleton, 2004; Suthers et al., 2011; Chapman et al., 2024) rather than direct control by winds. Topographic interaction could play a role, particularly during the inshore mode, however, further investigation is needed.

We also observe a shoaling of the nutricline (the maximum absolute vertical gradient in nutrient concentrations) during the offshore mode compared to the inshore mode (Figure 11). This effect is observable in Figure 9 as the inshore mode has a low vertical gradient across the majority of the transect, compared to the offshore mode, which has a sharper gradient in nutrients observable within the top 200 dbar. This effect of a shoaling nutricline was also observed in the Kuroshio "large meander" mode, which combined with winter convective mixing, made nutrients accessible to the surface waters (Hayashida et al., 2023). However, this area within the upstream EAC does not experience strong winter convective mixing and is characterised by relatively small seasonal changes in MLD. We found a maximum mean MLD of 40 m in winter, similar to what has previously been observed (Sobral et al., 2024; Condie and Dunn, 2006). Previous studies found minimum MLDs in summer of 20 m (Sobral et al., 2024) and 30 m (Condie and Dunn, 2006). This indicates that the MLD changes by only 10–20 m throughout the year. As such, convective mixing has low potential for replenishing mixed layer nutrients. Instead, we find that the properties of the water column are primarily being influenced by EAC variability.

The Southwest Pacific surface waters are oligotrophic, and plankton in this area are nitrate and phosphate limited (Ellwood et al., 2013; Ustick et al., 2021). The upwelling of nutrient rich waters, observed in this study, can cause plankton blooms (Silsbe and Malkin, 2016; Chapman et al., 2024) and has implications for ecosystem function (Brander et al., 2003; Hays et al., 2005; McGillicuddy Jr, 2016; Richardson and Schoeman, 2004). For example, data from a single voyage (*IN2019_V05* from Table 1) showed uplift of deeper waters and upwelling velocities along the strongly tilted isopycnals which form the inshore flank of the EAC (Chapman et al., 2024). When the EAC was in the offshore position, this uplift was associated with surface phytoplankton blooms and increased zooplankton biomass (Chapman et al., 2024). Whilst we have observed mechanisms which lead to an increase of nutrients in the upper water column, exploring biological responses to such nutrient changes is out of the scope of this paper. Future work should explore these relationships further with sampling and high-resolution biogeochemical models.

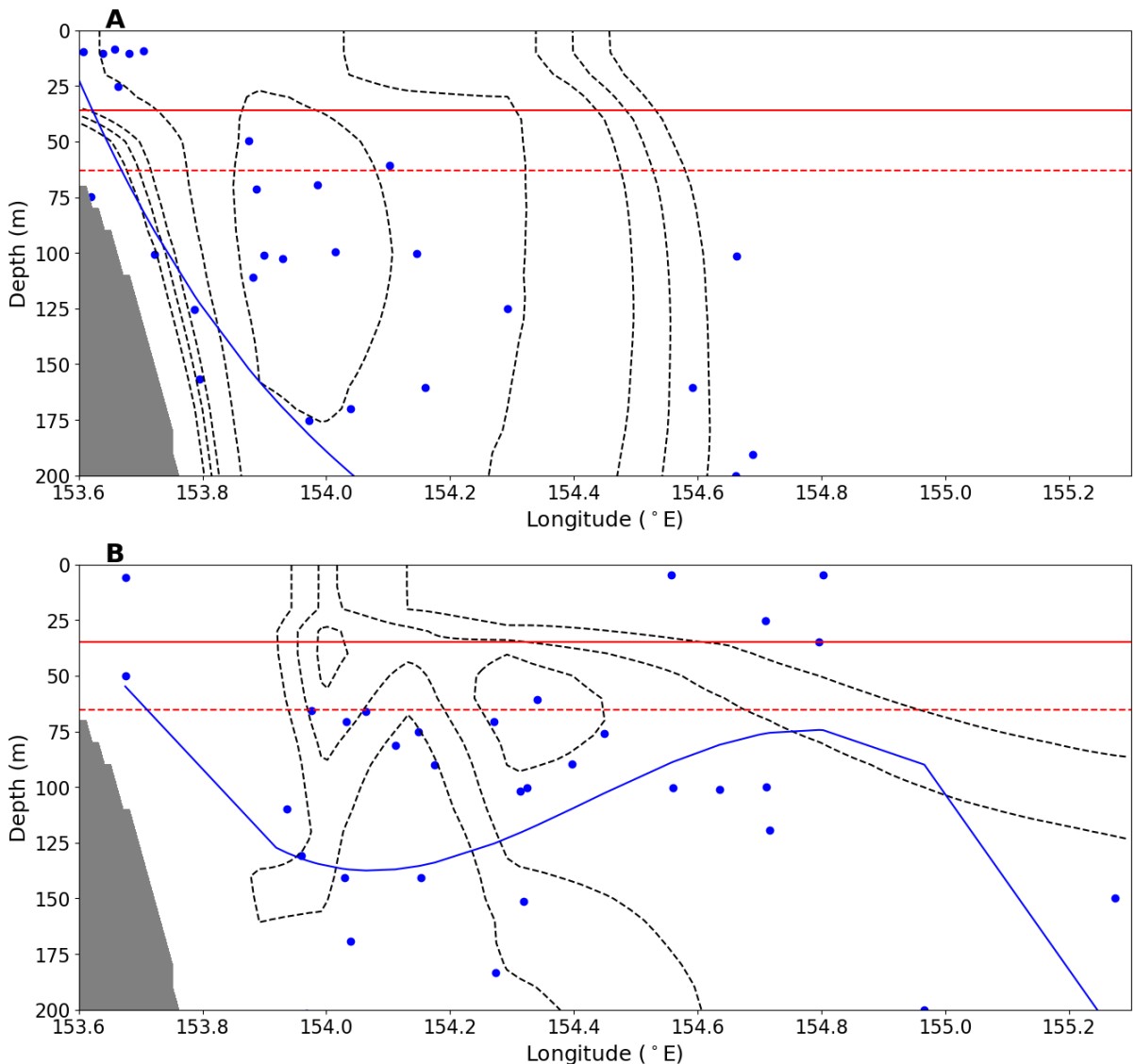

**Figure 11.** Nutricline depth (dots) for CTD casts for (a) inshore and (b) offshore mode. Nutricline depth was calculated as the maximum vertical gradient of full-depth bottle samples for nitrate. Blue line is polynomial robust line of fit for calculated nutricline depths. A lowess line of fit was also plotted and agreed well with the robust line of fit. Dashed black lines show the 0.6, 0.4, 0.25, 0.2, and 0.15 $m/s$ contours of mean southward velocity for each corresponding mode. Bold red line shows mean mixed layer depth and dashed red line shows 90[th] percentile mixed layer depth. Grey shading represents the seafloor.

The improved understanding of biogeochemical dynamics and the associated plankton response will improve our ability to understand how marine ecosystems may respond to future variability in the EAC.

## 5    Conclusions

This study systematically assesses the nutrient variability in the EAC intensification region, and is one of the few studies focusing on waters beyond the coastal and shelf environment. We find seasonal variability in the temperature, salinity, oxygen, and nutrients of the EAC, and importantly we identify the influence of EAC jet meandering on the uplift of nutrients. We categorised the EAC into two distinct modes based on the position of the current relative to the continental shelf and slope — the inshore and offshore modes. The inshore mode is generally low in nutrients, with high nutrient waters trapped and concentrated between the EAC jet the continental slope. The offshore mode, when the EAC jet sits over the abyssal plain, exhibits a shoaled nutricline, with nutrient-enriched waters reaching the mixed layer in the offshore region. Whilst we cannot attribute these changes to any one previous hypothesis of EAC variability, we see that the position and strength of the EAC influences the supply of nutrients to the upper ocean, likely due to oceanic processes such as eddy interactions. Understanding the response of nutrients to the movement of the EAC jet contributes to our knowledge of the EAC's role in supporting and shaping the biological communities and productivity within this region.

*Author contributions.*  BS and CC planned and led voyages and supplied data. MJ undertook the analysis and prepared the original paper draft with contributions from CC, BS and HB. All authors contributed to preparation of the final published paper.

*Competing interests.*  At least one of the (co-)authors is a member of the editorial board of Ocean Science.

*Acknowledgements.*  We acknowledge the use of the CSIRO Marine National Facility (https://ror.org/01mae9353) and grant of sea time on RV *Investigator* in undertaking this research. CSIRO Environment and IMOS funded the collection of the EAC mooring array. IMOS is enabled by the National Collaborative Research Infrastructure Strategy (NCRIS). CSIRO Environment funds were used to create the EAC data products. The authors would also like to acknowledge the hydro-chemistry teams from all voyages that undertook nutrient analysis, and the ship's crew, officers and MNF support staff for their support during various field campaigns .

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
