# Peer review of "Control of spatio-temporal variability of ocean nutrients in the East Australian Current"

_EGUsphere, 2024_

## Author Comment (AC1)

**Major Comments**

- *I particularly liked the well-referenced introduction, clearly outlining our current understanding of the system both in terms of physical and biogeochemical oceanography. Between lines 49 and 64, the authors present various published hypotheses on the drivers behind the nutrient variability. I feel like this is exactly what the current study can help elucidate. However, I don't think you make it that clear in the conclusion which of the previous hypotheses better fits with what your data showed, or if there is a different new finding. I just think a stronger ending without leaving so many open questions would benefit the paper.*

Thank you for your encouraging remarks. The hypotheses presented in the introduction will be revisited in the discussion to contextualize the nutrient variability. While we may not be able to attribute variability to one driver, with our planned additional analyses, we will make efforts to clarify our findings and provide a more robust conclusion

**Particular Comments**

- *In lines 68-71, you are presenting results from the current study in the Introduction section. It's good to mention what you will be doing without revealing the findings.*

We have considered the reviewers comments. We have revised the final paragraph of the introduction to now provide a synthesis of the major conclusions reached in this study.

- *Figure 1 could benefit from an inset zoomed-out map to show where exactly on the Australian coast the study area is located. I think you have space for this in the bottom-left corner.*

We will make several enhancements to Figure 1, including adding a subplot with surface currents and the location of the study area on the Australian coastline, providing additional context for readers unfamiliar with the region.

- *Figure 1 legend – the triangles are yellow, no?*

Fixed.

- *Section 2.2 You already have a section called "Methods" – I don't think you can have another subsection with the same name*

We have changed the subheading to "Analysis".

- *Line 110 – You give the citation for MLD calculation. Can you be more specific and explicitly state in the text what this means (e.g. density difference from surface)?*

We used the Holte and Talley (2009) algorithm to calculate the mixed layer depth (MLD). This method uses an algorithm to determine the MLD from a suite of possible MLDs based on various methods (density and temperature difference from the surface, and temperature and density gradient criteria). The threshold value of temperature and density for both the difference and gradient methods are 0.2°C and 0.03 kg m$^{-3}$, and 0.005°C dbar$^{-1}$ and 0.0005 kg m$^{-3}$ dbar$^{-1}$, respectively. We will add further information on the application of the Holte and Talley (2009) algorithm to determine the MLD in the revised manuscript.

- *Figure 2 – Is this the full 10 years of data? Maybe make this clear to the reader? Also, for example, at 154.8, I see two lines in the CTD plot (green and yellow), but I only see green dots in the plot above. Does this mean no samples were collected from the CTD on one occasion? Please clarify. Furthermore, you need units on your axis labels.*

We will update the figure and caption to improve the clarity. Figure 2 includes the complete 10 years of data. The hydrochemical plot appears sparser because bottle samples were not collected at all CTD casts. We will revise the figure caption and methods section appropriately. Additionally, the Figure 2 axis labels have been updated to include the appropriate units.

- *Line 118 and Figure 3 – How come? Is the EAC mostly (a majority of the total time; or a majority of the time samples) in the inshore mode making the average being so close to the inshore plot?*

Yes, the reviewer is correct: the inshore mode, where the EAC flows over the continental slope, is the climatological mean position of the EAC. In our dataset, the inshore mode occurs for 57% of samples. We have provided further information regarding the inshore mode alignment with the climatological mean EAC position in the manuscript.

- *Figure 4 – I find myself confused about what the stippling means. In this caption you say it indicates statistically significant departures from the average, but in the methods section you say that stippling shows areas of statistical significance.*

We will edit the two descriptions for consistency.

- *Figure 5 could benefit from three labels that show what is shown on each horizontal row: autumn, winter, spring. Figure 5 caption should say "light grey lines".*

Each subplot has been updated to include the corresponding season in its description. Additionally, the caption has been revised to specify "black dotted lines" to match changes to the figure.

- *Line 178: "there are a few significant changes"*

Fixed.

- *Figure 8: I think you need to explain to the reader what a violin plot is and think about if it's really useful to show the data in this way. Even in the text, you only very briefly mention this figure but you don't interpret it almost at all. Is there a significance of the flat base of some violins in subfigures b and c being higher than 0?*

Thank you for your insightful comments. After careful consideration, we have modified the text to provide a comprehensive interpretation of what is shown by this figure. In addition, we decided to replace the violin plots with boxplots, which provide a clearer representation of the data and more appropriately match the interpretations in the text.

- *Line 204: "Interesting impact" is too subjective.*

We have edited to remove subjectivity.

- *Line 205 and line 211: don't these two statements contradict each other? Are the water properties changed or not between the different modes?*

While the overall ranges of water properties remain unchanged when geographical data is excluded, clear differences emerge when geographical data and depth are included in the analysis. We have revised the text for clarity to better convey this distinction.

- *Line 207: "with of cooler"? something missing?*

Fixed.

- *Lines 229 and further: while I appreciate putting your results in the context of other studies, I feel like you are drifting too far from your results and instead it turns into a literature review. Please base your discussion more on your findings.*

We will relocate the discussion on WBCs to the introduction. The discussion will now retain only a few relevant remarks that are directly compared to our findings, ensuring a more focused analysis.

- *Line 247: That is exactly what you should find out through a study like this one, no?*

The intent was to highlight that many questions remain unanswered. The text will be altered for clarity.

- *Line 256: data are needed*

This section will be removed.

- *Final remarks: You have 10 years of data to work with. While probably not long enough to find statistically significant biogeochemical trends, should you not mention if you have seen any trends at all? Maybe a timeline plot, or a Hovmöller where you have longitude and time on your axes at a chosen depth?*

We would have very much liked to investigate long-term trends in nutrients in the region, and this project started with that idea in mind. Unfortunately, the data do not permit robust investigation of trends. Although we have 10 years of data, the sampling frequency is quite sparse, occurring approximately every 18 months. This results in effectively only seven sampling periods, which are collected at different times of the year and are subject to large seasonal and interannual variability.

---

## Author Comment (AC2)

**Major Comments**

- *1) The near-shore nutrient-rich water west of the EAC jet has been attributed to upwelling (L272), but I am not fully convinced that it is the cause because it is not shown. Can this be shown using data below 200 m (L12 refers to >200 m, but the results presented are restricted to the upper 200 m)? Or is this attribution a speculation based on previous knowledge? In the case of the Kuroshio (Hayashida et al. 2023), it was attributed to the uplifting of nutricline that makes sub-surface nutrient-rich water accessible to the surface via winter convective mixing.*

We appreciate the reviewer's insightful comment. After reviewing Hayashida et al. (2023), we will be including an analysis of nutricline depth in our manuscript. Some initial analysis shows that similar to their findings, we observe a shoaling of the nutricline during the offshore mode, with mean nutricline depths of approximately 350 m in the inshore mode and 200 m in the offshore mode. While our dataset doesn't allow analysis of seasonal MLD variability, we will use observed gridded products and reanalysis products to investigate the dynamics that results in nutricline depth variability and determine whether convective mixing plays a role. We believe this will greatly improve our manuscript.

*2) The findings about the inshore and offshore modes (L12-14) need to be supported by improved figures. The transect figures (Figures 4,6,7,9) show the inshore and offshore composites of nutrients but none of these show the corresponding locations of the EAC jet, which makes it difficult to understand the relationship between the EAC jet position and the nutrient distributions. Would it be possible to add the locations of the inshore and offshore EAC jets based on composites to these figures? Furthermore, while Table 2 is useful for understanding the probability distribution of inshore vs offshore modes, it would be also useful to produce time series of the longitudinal location of the EAC jet (similar to the Kuroshio axis time series such as Figure 2a of Hayashida et al. 2023). This will help better understand the temporal variations in the EAC jet and link with climate variability such as ENSO (L250).*

We will be incorporating composite velocity contours into the transect figures and add a time series of the EAC jet position (for sampled days) to Figure 3. However, we have opted to remove the discussion of ENSO from the manuscript. Given the sparsity of the data, we believe that additional time series may not provide any useful further insights.

- *3) Introduction contains a lot of useful background information on the jets that governs the EAC and extensions. Many numbers (latitudes) are mentioned, but it would be helpful to visualize these details. Figure 1 can be zoomed out to include these details with arrows and lines? It can also denote 154.2 E along the transect used for the jet position definition (L117).*

We will be making several revisions to Figure 1. We will retain the original scale of Figure 1, but it will now serve as a subplot to the figure. The second subplot will be zoomed out to show all currents mentioned in the introduction. Following the reviewer's suggestion, we will also mark the line for 154.2° E on the original Figure 1.

**Particular Comments**

- *L20-21: Consider citing a reference relevant to this sentence.*

We will add citations for reviews of the EAC and water masses/circulation of the southwest pacific.

- *L58: "coastal" is adjective. Suggest adding a word after it or use "coast" instead.*

Fixed.

- *L84: "intensification zone". It would be helpful to indicate this zone in an introductory figure.*

Instead of creating a separate introductory figure, we will add a subplot to Figure 1 that includes surface currents. Alongside the definition provided in the introduction, this addition should clarify the location of the intensification zone.

- *Figure 1: The caption says "black triangles" but they are yellow in the figure.*

Fixed.

- *L104: Suggest deleting "depth" at the end of the sentence.*

Fixed.

- *L106: This reasoning only supports the claim for phosphate limitation. Any reference for nitrate limitation?*

Yes, we will add references that examine nutrient ratios and utilize nutrient enrichment experiments to demonstrate nitrate limitation.

- *L107: This sentence is a bit awkward; silicate limitation could be due to large uptake of silicate by these diatoms. Suggest rephrasing something like "Siliceous diatoms dominate … that contributes to silicon limitation in this region".*

We will edit for clarity.

- *L112: Suggest adding a few words to briefly describe the density-based procedure.*

We will add a description to provide more context and further details regarding the methods employed in the Holte and Talley (2009) algorithm.

- *L113: "grouped by season". It is unclear which months are considered as which season. This could be indicated in Figure 2 caption. (\*I noticed these are mentioned later in L131, but they should be mentioned here at first appearance).*

Yes, we will add the months included in each season to the text and to the Figure 2 caption.

- *L114: "qualitative". I think this definition is rather quantitative. It seeks the longitude at which the southward velocity between 40-100 m is at maximum.*

We will edit for clarity.

- *L116: "was between 40-100 m" should be "between 40-100 m was" because the former sounds like the strongest flow can be located beyond the 40-100 m range, but the method looks for the strongest flow within the 40-100 m range, correct?*

Fixed.

- *L113: The definition of the inshore/offshore mode is a bit awkward and can be written more simply, something like "The EAC is considered as an inshore mode when the strongest southward flow between 40-100 m along the CTD transect is located west of 154.2E. Otherwise, it is considered an offshore mode".*

We will be editing this section to simplify the mode definition.

- *Figure 2: Markers (a,b) and lines (c,d) often overlap, which makes it difficult to visualize the density of the data coverage. Suggest fine-tuning the figure with the transparency parameter ("alpha" in Python) for markers and lines.*

Fixed.

- *Figure 3 b and c: Since the rest of the analysis is based on the inshore and offshore composites, would it be more consistent and robust to show the composites in these panels too, instead of an example transect?*

While this is a valid point, the significant variability in the offshore mode may complicate the interpretation of a composite, as there is no distinct EAC jet. After considering both options, we concluded that an example day provides clearer definitions of the modes. However, we have added the composite data to the transect figures (Figures 8 and 9) for additional context and are considering adding a second column to this figure with the composites.

- *L144: The past tense "was" is used twice, whereas the rest of the section uses present tense. Suggest changing "was" to "is" for consistency.*

Fixed.

- *Figure 4: The caption says "upper 200 m" but the y-axes are given as "pressure". Also, the widths of the left panels are narrower than those of the left and right panels as they are squeezed by the presence of the colorbars. It is also hard to read the longitudes as they are close to each other (some possible solutions are to: reduce the number of decimals displayed, tilt the labels, or use fewer ticks/labels). The same goes for the subsequent figures displaying the vertical distributions.*

All points here will be addressed in revised figures.

- *Figure 5: It took me a while to figure out what the grey circles represent. It is a bit misleading to label panel A as autumn, but it shows for other seasons too (and the same goes for other panels). While I understand the benefit of showing all data in grey as background, these panels (A/C/E and B/D/F) are easily comparable without the grey circles because they are shown using the same x and y axes. Therefore, I would suggest removing the grey circles for potential confusion. Also, the caption refers to the isopycnal lines as "light grey", but they look more like black. Suggest referring to these lines as "black dotted lines".*

We respectfully disagree on this point. Although the panels are adjacent, we find that including the grey points representing "all data" makes it easier to clearly identify changes between seasons in the TS plots, providing valuable context for comparison. This is also common practise in the literature. We will add further clarification in the figure caption to mitigate any confusion.

- *Figure 6: Why does the colorbar for nitrate include negative values? Also, the colors range from light to dark for nitrate and phosphate, whereas it goes from dark to light for silicate, which makes the visual comparison counterintuitive. The same goes for Figure 9.*

We will adjust all colorbars.

- *L155: Figure 8 is cited before Figure 7. In this case, the order of these figures should be switched.*

Yes, Figure 7 and 8 will be switched.

- *L157-159: It is unclear whether these sentences are referring to Figure 8 or Figure 6.*

This refers to Figure 8. We will add additional figure references for clarification.

- *L167 and L170: Missing the closing brackets for the figure citations.*

Fixed.

- *L182: "was" should be "is" for consistency with the rest of the text in the section?*

Fixed.

- *L184-L186: Suggest deleting this paragraph, because it is based on the results not shown and also because part of the results is already mentioned in L175-L177 and the range can be inferred from Figure 7.*

We respectfully disagree with the reviewer on this point. We find this to be useful information, however we will significantly shorten the paragraph and combined it with the description of Figure 8 for better integration with the relevant results.

- *Figure 8: It would be helpful to provide a description for violin plots, as I think not all Ocean Science readers are familiar with violin plots, which seems more complex than others like Taylor diagrams or box plots. At the least, please provide the reference where readers can obtain the necessary information to understand these plots.*

After consideration, we have decided to go a different route and change the violin plot to boxplot, which more clearly represents the data and matches the interpretations in the text. We will modify the text appropriately.

- *L205: "the average properties of the water column" is unclear. Which properties (temperature?) and what does it mean by "average"?*

This will be edited.

- *L207: "we see evidence of upwelling" is unclear. Was such evidence shown in Results?*

This section will be rewritten to clarify our point more effectively.

- *L210: "meaning that upwelling in the offshore mode results in higher ..." requires information on the vertical location of the nutrient rich water, rather than the horizontal location mentioned in the previous sentence.*

Similarly to the previous point, this section will be rewritten.

- *L249: I am not sure if it is ok to bring the not-shown-results into discussion. Is there a reason for not showing the results in the paper? I think it would be beneficial to show such time series comparison even though the lack of temporal coverage. If page limit is an issue, it can be included as supplementary information.*

We have opted to delete this section entirely, as it diverges too far from the main message of the manuscript.

- L272: ". M" should be ", m"

Fixed.

---

## Author Response (AR1)

**Response to Reviewer Comments on *Jeffers et al*: Control of spatio-temporal variability of ocean nutrients in the East Australian Current**

Megan Jeffers, Christopher Chapman, Bernadette Sloyan, Helen Bostock

October 2024

We thank the reviewers for the thorough review of our paper and providing many constructive comments. We have addressed all comments and believe this has resulted in an improved paper.

Below, we present the response to each reviewer comment. Original reviewer comments are in italics, our response is in normal text, and excerpts from the revised manuscript are in blue.

**1 Response to Reviewer #1**

We have addressed all minor comments and have carefully considered major comments and suggestions for improvement.

**1.1 Major Comments**

- *I particularly liked the well-referenced introduction, clearly outlining our current understanding of the system both in terms of physical and biogeochemical oceanography. Between lines 49 and 64, the authors present various published hypotheses on the drivers behind the nutrient variability. I feel like this is exactly what the current study can help elucidate. However, I don't think you make it that clear in the conclusion which of the previous hypotheses better fits with what your data showed, or if there is a different new finding. I just think a stronger ending without leaving so many open questions would benefit the paper.*

Thank you for your encouraging remarks. We have made efforts to clarify our findings and provide a more robust conclusion. The hypotheses presented in the introduction have been revisited in the discussion to contextualize the nutrient variability. However, in this study we don't think that any one hypothesis can definitively describe the mechanism driving the nutrient variability. The study finds that the offshore EAC jet is often associated with a inshore cyclonic eddy. Chapman et al. (2024) and other studies find that inshore cyclonic eddies result in upwelling driven by internal ocean processes. Topographic interactions at the continental slope may play a role in upwelling of nutrients when the EAC is located at its inshore mode. However, further work is needed to understand the EAC-topographic interactions.

We have added comments to the Discussion and Conclusion sections:

The discussion now reads:

In the process of classifying EAC modes for this study, it was noted that the offshore mode is often associated with a cyclonic eddy sitting on the inshore side of the EAC, similar to what was observed by Chapman et al. (2024). As such, upwelling is likely linked to oceanic internal processes, such as eddy interactions and sub-mesoscale circulations (*e.g.*, Roughan et al., 2017; Roughan and Middleton, 2004; Suthers et al., 2011; Chapman et al., 2024) rather than direct control by winds. Topographic interaction could play a role, particularly during the inshore mode, however, further investigation is needed.

And the conclusion now reads:

Whilst we cannot attribute these changes to any one previous hypothesis of EAC variability, we see that the position and strength of the EAC influences the supply of nutrients to the upper ocean, likely due to oceanic processes such as eddy interactions. Understanding the response of nutrients to the movement of the EAC jet contributes to our knowledge of the EAC's role in supporting and shaping the biological communities and productivity within this region.

**1.2 Particular Comments**

- *In lines 68-71, you are presenting results from the current study in the Introduction section. It's good to mention what you will be doing without revealing the findings.*

  We have considered the reviewers comments. We have revised the final paragraph of the introduction to now provide a synthesis of the major conclusions reach in this study.

  The Introduction final paragraph includes the following:

  We examine the seasonality of the nutrients and the role of the position of the EAC, relative to the continental shelf and slope, in influencing the distribution of nutrients in the upper water column. Our analysis highlights the important interactions between nutrient concentrations and distribution and the highly variable EAC, which has implications for primary production, fisheries, and the biological carbon pump. Understanding the dynamical implications of the position of EAC on nutrient distribution is essential for elucidating the broader implications of the EAC's role on marine ecosystems and fisheries. This long-term dataset offers a valuable insight into the EAC's dynamical influence on the surface and mixed layer oceanography, biogeochemical cycling, and nutrient concentrations.

- *Figure 1 could benefit from an inset zoomed-out map to show where exactly on the Australian coast the study area is located. I think you have space for this in the bottom-left corner.*

  We have made several enhancements to Figure 1. The figure now includes a panel displaying surface currents along with the study location on the Australian coastline, providing additional context for readers unfamiliar with the region. The original Figure 1 has been repositioned as the second panel.

- *Figure 1 legend – the triangles are yellow, no?*

  Fixed.

  The revised Figure 1 and caption is:

[Figure]

Figure 1: Figure 1: Location of data in context of broader surroundings. Panel (a) shows 20-year time mean (Jan 2000 – Dec 2019) of surface geostrophic currents from IMOS OceanCurrent. The location of the EAC mooring array is indicated by the red line. The black star shows the location of Brisbane, Australia and NC=New Caledonia. The black square outlines the region panel (b). Panel (b) shows the location of CTD stations (red dots) and EAC array moorings (yellow triangles) used in this study. The bold white line shows the position of the 154.2°E line used for defining EAC modes. The bathymetric map of the study area is also shown with the contour interval (black line) every 500 m from 500–4000 m water depth.

- *Section 2.2 You already have a section called "Methods" – I don't think you can have another subsection with the same name*

We have changed the subheading to "Analysis".

- *Line 110 – You give the citation for MLD calculation. Can you be more specific and explicitly state in the text what this means (e.g. density difference from surface)?*

We used the Holte and Talley (2009) algorithm to calculate the mixed layer depth (MLD). This method uses an algorithm to determine the MLD from a suite of possible MLDs based on various methods (density and temperate difference from the surface and temperature and density gradient criteria). The threshold value of temperature and density for both the difference and gradient methods are 0.2°C and 0.03 kg m$^{-3}$, and 0.005°C dbar$^{-1}$ and 0.0005 kg m$^{-3}$ dbar$^{-1}$, respectively. We have added further information on the application of the Holte and Talley (2009) algorithm to determine the MLD in the revised manuscript.

The manuscript reads:

Mixed Layer depth (MLD) was calculated using density-based procedures developed by Holte and Talley (2009). This method uses an algorithm to choose the MLD from a suite of possible MLDs which are calculated using multiple threshold (difference from the surface) and gradient (where the depth-gradient exceeds a criteria) criteria. The threshold value of temperature and density for both the difference and gradient methods are 0.2°C and 0.03 kg m$^{-3}$, and 0.005°C dbar$^{-1}$ and 0.0005 kg m$^{-3}$ dbar$^{-1}$, respectively.

- *Figure 2 – Is this the full 10 years of data? Maybe make this clear to the reader? Also, for example, at 154.8, I see two lines in the CTD plot (green and yellow), but I only see green dots in the plot above. Does this mean no samples were collected from the CTD on one occasion? Please clarify. Furthermore, you need units on your axis labels.*

We have updated the figure 2 and caption, and Table 1. Figure 2 includes the complete 10 years of data. The hydrochemical plot has sparser data coverage because bottle samples were not collected on all CTD casts. We have revised the figure and Table captions and

methods section appropriately. Additionally, the Figure 2 axis labels have been updated to include the appropriate units.

The Figure 2 caption now reads:

Full 10 years of hydrograhic data points (coloured dots and lines) and grid (black lines) used for interpolated property sections. Distribution of nutrient data (a) grouped by season (austral autumn (March–May), winter (June–August), and spring (September–November)), and (b) EAC mode, and distribution of CTD data (c) grouped by seasons, and (d) EAC mode.

Table 1 now includes the number of CTD casts that have nutrient data.

And the manuscript (Section 2.1.2) now includes the following:

For the assessment of the EAC's physical and biogeochemical properties, we utilise 162 Conductivity, Temperature, Depth (CTD) and Niskin bottle profiles collected during research voyages (Figure 1, Table 1). Of the 162 CTD stations, nutrient samples were collected on 136. CTD profiles provide measurements of temperature, salinity, and dissolved oxygen at 1 dbar pressure intervals. Water samples for nutrient analsyes were collected by Niskin bottles at discrete depths that span the entire water column.

- *Line 118 and Figure 3 – How come? Is the EAC mostly (a majority of the total time; or a majority of the time samples) in the inshore mode making the average being so close to the inshore plot?*

Yes, the reviewer is correct: the inshore mode, where the EAC flows over the continental slope, is the climatological mean position of the EAC. In our dataset, the inshore mode occurs for 57% of samples. We have provided further information regarding the inshore mode alignment with the climatological mean EAC position and revised Figure 3.

We have modified Section 2 as follows

The EAC is considered to be in an inshore mode when the EAC core is located westward of 154.2°E (approximately over the continental slope, Figure 1 (b), bold white line), other-

[Figure]

Figure 2: Figure 3: Panel (a) shows the cumulative sum of days observed for the inshore (blue) and offshore (purple) modes, considering only when a CTD station was occupied. Meridional velocity ($m/s$; positive is equatorward) plotted for the upper 500 m of the water column from the daily EAC mooring data product for (b) mean over all days with a CTD station. A one-day example for (c) inshore mode, and (d) offshore mode. Also plotted is mean EAC jet for all days classified as (e) inshore mode, and (f) offshore mode.

wise it is considered an offshore mode. Comparing the velocity structure of the "inshore" and "offshore" modes with Sloyan et al. (2016), we note that the inshore mode essentially corresponds to the climatological southward velocities, which we have confirmed by taking composite averages of the meridional velocity across all days where a CTD station was occupied and where time steps are classified as inshore mode (Figure 3 (b) and (e) respectively). For the time period considered here, 57% of samples are categorised as belonging to the inshore mode. In contrast, the "offshore mode" approximately matches mode 2 from Sloyan et al. (2016) and diverges from the climatological position of the EAC.

- *Figure 4 – I find myself confused about what the stippling means. In this caption you say it indicates statistically significant departures from the average, but in the methods section*

*you say that stippling shows areas of statistical significance.*

We have edited the two descriptions for consistency.

The methods section reads:

To test for statistical significance a Monte Carlo simulation was run. For each variable, a random subset of data points were selected and used to make an interpolated pressure-longitude transect. This was repeated 1000 times and sections were constructed by selecting only the $<5^{\text{th}}$ and $>95^{\text{th}}$ percentile observations for each grid cell from the Monte Carlo. Stippling was added to all seasonal and mode sections for areas of statistical significance at the 95% confidence interval.

And the figure caption reads:

Stippling indicates statistical significance following the procedure described in the Analysis section (section 2.2).

- *Figure 5 could benefit from three labels that show what is shown on each horizontal row: autumn, winter, spring. Figure 5 caption should say "light grey lines".*

  Each subplot has been updated to include the corresponding season in its description. Additionally, the caption has been revised to specify "black dotted lines."

- *Line 178: "there are a few significant changes"*

  Fixed.

- *Figure 8: I think you need to explain to the reader what a violin plot is and think about if it's really useful to show the data in this way. Even in the text, you only very briefly mention this figure but you don't interpret it almost at all. Is there a significance of the flat base of some violins in subfigures b and c being higher than 0?*

  Thank you for your insightful comments. After careful consideration, we have modified the text to provide a comprehensive interpretation of what is shown by this figure. In

[Figure]

Figure 3: Box plots of nutrient concentrations across (a), (b), (c) the three seasons (Austral autumn, winter, and spring) and (d), (e), (f) the two modes sampled. Coloured dots are data points. Coloured boxes represent interquartile range (IQR), with the line inside the box representing the median (50th percentile). The whiskers (vertical black line) represents the range that contains 1.5 times the IQR above or below the first and third quartile.

addition, we decided to replace the violin plots with boxplots (see below), which provide a clearer representation of the data and more appropriately match the interpretations in the text.

We have elaborated slightly on the boxplots, with the seasonal results section now reading:

There is evidence of a subtle seasonal cycle in nutrients in the upper 200 dbar of the water column, as shown in Figure 7 (a)–(c). Within each season, there is a large degree of variability. In general, autumn and spring have similar IQR, and winter has the smallest IQR for all nutrients sampled. This suggests winter has the lowest nutrient concentrations available to the surface waters compared to the other seasons. Changes are broadly consistent across the three different nutrients, particularly in the variance and IQR. Although broadly similar, there are differences in the mean nutrient concentrations between

the three nutrients with austral season. There is little change to the nitrate mean, although there is an increased number of observations with high concentration in autumn and spring compared to winter (Figure 7 (a)). Phosphate experiences a slight increase in the seasonal mean from autumn through to spring (Figure 7 (b)). Silicate has the largest mean in autumn and winter and the lowest mean in spring (Figure 7 (c)).

And the modes section now reading:

When considering changes in nutrient concentrations for all data in the upper 200 dbar of the water column, there is no clear change in mean nutrient concentrations between the two modes, although the offshore mode has a larger IQR for all three nutrients sampled (Figure 7 (d)–(f)). The larger IQR in the offshore mode results from all nutrients having greater variability, reflecting a larger amount of samples with high nutrient concentration. This suggests a greater availability of nutrients in the upper 200 dbar when compared with the inshore mode. Similar to the changes across the three seasons, changes across the two modes are similar between the three nutrients. Additionally, the differences in nutrient concentrations between the two modes are on the same order of magnitude as seasonal changes.

- *Line 204: "Interesting impact" is too subjective.*

We have edited to remove subjectivity. This line now reads:

While we observe a seasonal nutrient signal, we also highlight the influence of the EAC core's position relative to the continental shelf/slope.

- *Line 205 and line 211: don't these two statements contradict each other? Are the water properties changed or not between the different modes?*

We have considered this comment. While the overall ranges of water properties remain unchanged when geographical data is excluded, clear differences emerge when geographical data and depth are included in the analysis. We have thoroughly revised the discussion, and in place of this line, the text now reads:

The position of the EAC affects the vertical distribution of nutrients and their longitudinal gradients along the transect.

- *Line 207: "with of cooler"? something missing?*

Fixed.

- *Lines 229 and further: while I appreciate putting your results in the context of other studies, I feel like you are drifting too far from your results and instead it turns into a literature review. Please base your discussion more on your findings.*

We have moved the discussion on WBCs to the introduction. The discussion now retains only a few relevant remarks that are directly compared to our findings, ensuring a more focused discussion section.

The introduction now reads:

Similar variability and jet meandering has been observed in other WBCs. The Kuroshio Current experiences variability in its core position and strength, and experiences meandering and interactions with bathymetry and mesoscale eddies (Ebuchi and Hanawa, 2003; Kawabe, 2005, 1995; Waseda et al., 2003). The Florida Current is the upstream portion of the Gulf Stream, and has a very similar jet structure to the EAC (Archer et al., 2018), and exhibits a similar meandering behaviour, with meanders occurring on a time-scale of 3–30 days (Archer et al., 2017). Other WBCs also experience meandering, including the Gulf Stream (Mao et al., 2023), Agulhas Current (Goschen et al., 2015), and the Brazil Current (Da Silveira et al., 2008).

Such current variability has been linked to biogeochemical variability in the EAC and other WBC systems. For example, in the EAC, Chapman et al. (2024) showed observations of nutrient injection into the surface layers during meandering, and that the vertical velocities that drove this were catalysed by the interaction of EAC with mesoscale eddies. Similarly, in the Kuroshio, the large meander mode results in an uplift of the nutricline, which increases nutrient availability in the near surface waters (Hayashida et al., 2023). In

the upstream region of the Kuroshio, increased speeds of the current causes an uplift of nutrients onshore (Chen et al., 2022). Additionally, during periods where the Kuroshio sits closer to the coast, it interacts with bathymetry which causes strong vertical mixing and uplift of nutrients to the continental shelf (Durán Gómez and Nagai, 2022). Mesoscale variability in the Florida Current is linked to the upwelling of cool, nutrient rich waters between the shelf break and the offshore meander (Fiechter and Mooers, 2007; Kourafalou and Kang, 2012). Upwelling has also been linked to the movement of jet meanders in the Agulhas Current, with upwelling occurring when the current shifts onto the continental shelf, or shifts offshore (Goschen et al., 2015). It is clear that meandering of WBCs upstream of their separation points is not unique to the EAC, and the current meandering can result in upwelling, which has been observed in several WBC systems.

And the Discussion now reads:

We also observe a shoaling of the nutricline (the maximum absolute vertical gradient in nutrient concentrations) during the offshore mode compared to the inshore mode (Figure 11). This effect is observable in Figure 9 as the inshore mode has a low vertical gradient across the majority of the transect, compared to the offshore mode, which has a sharper gradient in nutrients observable within the top 200 dbar. This effect of a shoaling nutricline was also observed in the Kuroshio "large meander" mode, which combined with winter convective mixing, made nutrients accessible to the surface waters (Hayashida et al., 2023). However, this area within the upstream EAC does not experience strong winter convective mixing and is characterised by relatively small seasonal changes in MLD. We found a maximum mean MLD of 40 m in winter, similar to what has previously been observed (Sobral et al., 2024; Condie and Dunn, 2006). Previous studies found minimum MLDs in summer of 20 m (Sobral et al., 2024) and 30 m (Condie and Dunn, 2006). This indicates that the MLD changes by only 10–20 m throughout the year. As such, convective mixing has low potential for replenishing mixed layer nutrients. Instead, we find that the properties of the water column are primarily being influenced by EAC variability.

The Southwest Pacific surface waters are oligotrophic, and plankton in this area are nitrate and phosphate limited (Ellwood et al., 2013; Ustick et al., 2021). The upwelling of nutrient

rich waters, observed in this study, can cause plankton blooms (Silsbe and Malkin, 2016; Chapman et al., 2024) and has implications for ecosystem function (Brander et al., 2003; Hays et al., 2005; McGillicuddy Jr, 2016; Richardson and Schoeman, 2004). For example, data from a single voyage (*IN2019_V05* from Table 1) showed uplift of deeper waters and upwelling velocities along the strongly tilted isopycnals which form the inshore flank of the EAC (Chapman et al., 2024). When the EAC was in the offshore position, this uplift was associated with surface phytoplankton blooms and increased zooplankton biomass (Chapman et al., 2024). Whilst we have observed mechanisms which lead to an increase of nutrients in the upper water column, exploring biological responses to such nutrient changes is out of the scope of this paper. Future work should explore these relationships further with sampling and high-resolution biogeochemical models. The improved understanding of biogeochemical dynamics and the associated plankton response will improve our ability to understand how marine ecosystems may respond to future variability in the EAC.

- *Line 247: That is exactly what you should find out through a study like this one, no?*

  The intent was to highlight that many questions remain unanswered. However, this may not be the appropriate context or phrasing for such a point. This part of the text has been removed.

- *Line 256: data are needed*

  This section has now been removed.

- *Final remarks: You have 10 years of data to work with. While probably not long enough to find statistically significant biogeochemical trends, should you not mention if you have seen any trends at all? Maybe a timeline plot, or a Hovmöller where you have longitude and time on your axes at a chosen depth?*

  We would have very much liked to investigate long-term trends in nutrients in the region,

[Figure]

Figure 4: Nitrate concentration time-series showing all data points for the top 50 (red) and top 100 (orange) dbars of the water column.

and this project started with that idea in mind. Unfortunately, the data do not permit robust investigation of trends. Although we have 10 years of data, the sampling frequency is quite sparse, occurring approximately every 18 months. This results in effectively only seven sampling periods, which are collected at different times of the year and are subject to large seasonal and interannual variability.

To make this point more clearly, we have included a nitrate concentrations versus time plot for the top 50 (red) and top 100 (orange) dbars. Given this limited dataset and high variability, it not possible to determine robust trends.

**2 Response to Reviewer #2**

We thank the reviewer for the constructive and supportive comments. Below, we provide a summary of our responses to the three main suggestions, followed by more detailed response to particular comments.

1. We have further investigated the inshore-offshore difference and investigated nutricline depth. This additional analysis suggested that the depth of the nutricline may play a role in the EAC mode differences. The revised manuscript now includes this discussion.

2. We have enhanced the inshore-offshore figures by incorporating velocity contours of the current. Additionally, we have included an analysis of the differences in nutricline depth between the two modes.

3. We have added a new panel to Figure 1, which now includes introductory information to provide better context.

**2.1 Major Comments**

- *The near-shore nutrient-rich water west of the EAC jet has been attributed to upwelling (L272), but I am not fully convinced that it is the cause because it is not shown. Can this be shown using data below 200 m (L12 refers to ¿200 m, but the results presented are restricted to the upper 200 m)? Or is this attribution a speculation based on previous knowledge? In the case of the Kuroshio (Hayashida et al. 2023), it was attributed to the uplifting of nutricline that makes sub-surface nutrient-rich water accessible to the surface via winter convective mixing.*

We appreciate the reviewer's insightful comment. After reviewing Hayashida et al. (2023), we have included an analysis of nutricline depth in our manuscript. Similar to their findings, we observe a shoaling of the nutricline during the offshore mode. Our analysis indicates significant shoaling for all nutrients, particularly for nitrate and phosphate, with median nutricline depths for nitrate of 300 m in the inshore mode and around 120 m in the offshore mode. We are unable to diagnose convective mixing in our dataset, however we have added references to previous observational and modelling studies which highlight the small seasonal change in MLD in our study region, showing convective mixing in the offshore region is relatively weak (Sobral et al., 2024; Condie and Dunn, 2006).

Following the reviewers comments, we have revised the manuscript and now include discussion of the nutricline depth in association with the EAC mode position to provide further insights into the interplay between EAC location and nutrient distribution.

The Discussion section has been modified to include the following text

We also observe a shoaling of the nutricline (the maximum absolute vertical gradient in nutrient concentrations) during the offshore mode compared to the inshore mode (Figure 11).

[Figure]

Figure 5: Figure 11: Nutricline depth (dots) for CTD casts for (a) inshore and (b) offshore mode. Nutricline depth was calculated as the maximum vertical gradient of full-depth bottle samples for nitrate. Blue line is polynomial robust line of fit for calculated nutricline depths. A lowess line of fit was also plotted and agreed well with the robust line of fit. Dashed black lines show the 0.6, 0.4, 0.25, 0.2, and 0.15 $m/s$ contours of mean southward velocity for each corresponding mode. Bold red line shows mean mixed layer depth and dashed red line shows $90^{\text{th}}$ percentile mixed layer depth. Grey shading represents the seafloor.

This effect is observable in Figure 9 as the inshore mode has a low vertical gradient across the majority of the transect, compared to the offshore mode, which has a sharper gradient in nutrients observable within the top 200 dbar. This effect of a shoaling nutricline was also observed in the Kuroshio "large meander" mode, which combined with winter convective mixing, made nutrients accessible to the surface waters (Hayashida et al., 2023). However, this area within the upstream EAC does not experience strong winter convective mixing and is characterised by relatively small seasonal changes in MLD. We found a maximum mean MLD of 40 m in winter, similar to what has previously been observed (Sobral et al., 2024; Condie and Dunn, 2006). Previous studies found minimum MLDs in summer of 20 m (Sobral et al., 2024) and 30 m (Condie and Dunn, 2006). This indicates that the MLD changes by only 10–20 m throughout the year. As such, convective mixing has low potential for replenishing mixed layer nutrients. Instead, we find that the properties of the water column are primarily being influenced by EAC variability.

- *The findings about the inshore and offshore modes (L12-14) need to be supported by improved figures. The transect figures (Figures 4,6,7,9) show the inshore and offshore composites of nutrients but none of these show the corresponding locations of the EAC jet, which makes it difficult to understand the relationship between the EAC jet position and the nutrient distributions. Would it be possible to add the locations of the inshore and offshore EAC jets based on composites to these figures? Furthermore, while Table 2 is useful for understanding the probability distribution of inshore vs offshore modes, it would be also useful to produce time series of the longitudinal location of the EAC jet (similar to the Kuroshio axis time series such as Figure 2a of Hayashida et al. 2023). This will help better understand the temporal variations in the EAC jet and link with climate variability such as ENSO (L250).*

We have incorporated composite velocity contours into the transect figures and added a time series of the EAC jet position (for sampled days) to Figure 3. However, we have removed the discussion of ENSO from the manuscript. Given the sparsity of the data, we believe that additional time series may not provide any useful further insights.

To provide insights into temporal variability of the EAC, we have undertaken an analysis

[Figure]

Figure 6: Figure 10: The EAC velocity anomaly of the first two modes of the Complex Empirical Orthogonal Function (CEOF) analysis. For the leading two modes, the spatial maps of (a), (d) phase ($-\pi - \pi$) and (b), (e) amplitude and (c), (f) reconstructed velocity anomaly (m/s) are shown. The inshore mode approximately matches the CEOF mode 1 (a), (b), (c), which accounts for 65.4% of velocity anomaly. The CEOF mode 2 (d), (e), (f) fits the offshore mode, and accounts for 13.3% of velocity anomaly. The reconstructed velocity anomaly for each mode is determined from the spatial amplitude and phase maps.

of the 10-year velocity data from the mooring array. The complex empirical orthogonal function (CEOF) analysis of the southward velocity component shows that the onshore and offshore EAC modes explain greater than 78% of the velocity variance with periods of 120-60 days. This additional analysis shows that EAC dynamics are separate from seasonality large scale dynamics. We have added a figure showing the CEOF leading 2 modes.

- *Introduction contains a lot of useful background information on the jets that governs the EAC and extensions. Many numbers (latitudes) are mentioned, but it would be helpful to visualize these details. Figure 1 can be zoomed out to include these details with arrows and lines? It can also denote 154.2 E along the transect used for the jet position definition (L117).*

We have revised Figure 1 to enhance clarity. While the original figure retains its scale, we have added a second panel that is zoomed out to include the currents mentioned. Additionally, we have marked the line for 154.2°E on the original Figure 1, which is now

designated as panel (b).

**2.2 Particular Comments**

- *L20-21: Consider citing a reference relevant to this sentence.*

  We have added two citations for reviews of the EAC and water masses/circulation of the southwest pacific by Suthers et al. (2011) and Chiswell et al. (2015).

- *L58: "coastal" is adjective. Suggest adding a word after it or use "coast" instead.*

  Fixed.

- *L84: "intensification zone". It would be helpful to indicate this zone in an introductory figure.*

  Instead of creating a separate introductory figure, we have added a panel to Figure 1 that includes surface currents. Alongside the definition provided in the introduction, this addition should clarify the location of the intensification zone.

- *Figure 1: The caption says "black triangles" but they are yellow in the figure.*

  Fixed.

- *L104: Suggest deleting "depth" at the end of the sentence.*

  Fixed.

- *L106: This reasoning only supports the claim for phosphate limitation. Any reference for nitrate limitation?*

Yes, we have added references that examine nutrient ratios and utilize nutrient enrichment experiments to demonstrate nitrate limitation by Hassler et al. (2011) and Doblin et al. (2016).

This section now reads:

The Southwest Pacific contains a low dissolved phosphorus region centred around 28°S (Martiny et al., 2019). However, $NO_3$:$PO_4$ ratios show that nitrate is still the primary limiting nutrient (Hassler et al., 2011), particularly in the top 200 m (Doblin et al., 2016).

- *L107: This sentence is a bit awkward; silicate limitation could be due to large uptake of silicate by these diatoms. Suggest rephrasing something like "Siliceous diatoms dominate ... that contributes to silicon limitation in this region".*

We have reworded to hopefully be clearer with the line of reasoning.

Silicate is also a key nutrient in this region, as siliceous diatoms dominate the phytoplankton community (Eriksen et al., 2019; Thompson et al., 2009). However, like nitrate and phosphate, silicate is also limited in this region, and is experiencing a decline (Ellwood et al., 2013; Thompson et al., 2009).

- *L112: Suggest adding a few words to briefly describe the density-based procedure.*

We have added a description to provide more context and further details regarding the methods employed in the algorithm Holte and Talley (2009) algorithm. The methods now reads:

Mixed Layer depth (MLD) was calculated using density-based procedures developed by Holte and Talley (2009). This method uses an algorithm to choose the MLD from a suite of possible MLDs which are calculated using multiple threshold (difference from the surface) and gradient (where the depth-gradient exceeds a criteria) criteria. The threshold value of temperature and density for both the difference and gradient methods are 0.2°C and 0.03 kg m$^{-3}$, and 0.005°C dbar$^{-1}$ and 0.0005 kg m$^{-3}$ dbar$^{-1}$, respectively.

- *L113: "grouped by season". It is unclear which months are considered as which season. This could be indicated in Figure 2 caption. (\*I noticed these are mentioned later in L131, but they should be mentioned here at first appearance).*

Yes, we have added the months included in each season to the text and to the Figure 2 caption.

- *L114: "qualitative". I think this definition is rather quantitative. It seeks the longitude at which the southward velocity between 40-100 m is at maximum.*

We have removed the word qualitative.

- *L116: "was between 40-100 m" should be "between 40-100 m was" because the former sounds like the strongest flow can be located beyond the 40-100 m range, but the method looks for the strongest flow within the 40-100 m range, correct?*

Fixed.

- *L113: The definition of the inshore/offshore mode is a bit awkward and can be written more simply, something like "The EAC is considered as an inshore mode when the strongest southward flow between 40-100 m along the CTD transect is located west of 154.2E. Otherwise, it is considered an offshore mode".*

Following the reviewers suggestions we have simplified how we define inshore and offshore EAC position. The definition now reads:

For each day that a CTD station was occupied, we classified the EAC into an "inshore mode" or an "offshore mode" based on the meridional velocity profiles and position of the EAC core from the mooring data. The EAC is considered to be in an inshore mode when the EAC core is located westward of 154.2°E (approximately over the continental slope, Figure 1 (b), bold white line), otherwise it is considered an offshore mode. Comparing the velocity structure of the "inshore" and "offshore" modes with Sloyan et al. (2016), we

note that the inshore mode essentially corresponds to the climatological southward velocities, which we have confirmed by taking composite averages of the meridional velocity across all days where a CTD station was occupied and where time steps are classified as inshore mode (Figure 3 (b) and (e) respectively). For the time period considered here, 57% of samples are categorised as belonging to the inshore mode. In contrast, the "offshore mode" approximately matches mode 2 from Sloyan et al. (2016) and diverges from the climatological position of the EAC.

- *Figure 2: Markers (a,b) and lines (c,d) often overlap, which makes it difficult to visualize the density of the data coverage. Suggest fine-tuning the figure with the transparency parameter ("alpha" in Python) for markers and lines.*

  Fixed, we have increased the transparency.

- *Figure 3 b and c: Since the rest of the analysis is based on the inshore and offshore composites, would it be more consistent and robust to show the composites in these panels too, instead of an example transect?*

  While this is a valid point, the significant variability in the offshore mode may complicate the interpretation of a composite, as the variability within the mode means that there is no distinct EAC jet. After considering both options, we concluded that an example day provides clearer definitions of the modes, so we have kept example days for Figure 3, but have added panels showing the composites. We have also added the composite data to the transect figures (Figures 8 and 9) for additional context.

- *L144: The past tense "was" is used twice, whereas the rest of the section uses present tense. Suggest changing "was" to "is" for consistency.*

  Fixed.

- *Figure 4: The caption says "upper 200 m" but the y-axes are given as "pressure". Also, the widths of the left panels are narrower than those of the left and right panels as they are squeezed by the presence of the colorbars. It is also hard to read the longitudes as they are close to each other (some possible solutions are to: reduce the number of decimals displayed, tilt the labels, or use fewer ticks/labels). The same goes for the subsequent figures displaying the vertical distributions.*

We have addressed these issues: the captions have been corrected, and the widths of all panels have been adjusted to be uniform. The longitude labels have been rotated for the seasonal transects, while for the mode transects, we have ensured they are spaced out sufficiently and display fewer decimals.

- *Figure 5: It took me a while to figure out what the grey circles represent. It is a bit misleading to label panel A as autumn, but it shows for other seasons too (and the same goes for other panels). While I understand the benefit of showing all data in grey as background, these panels (A/C/E and B/D/F) are easily comparable without the grey circles because they are shown using the same x and y axes. Therefore, I would suggest removing the grey circles for potential confusion. Also, the caption refers to the isopycnal lines as "light grey", but they look more like black. Suggest referring to these lines as "black dotted lines".*

We respectfully disagree on this point. Although the panels are adjacent, we find that including the grey points representing "all data" makes it easier to clearly identify changes between seasons in the TS plots, providing valuable context for comparison. However, to address the reviewers comments, we have revised the caption to minimise any confusion.

- *Figure 6: Why does the colorbar for nitrate include negative values? Also, the colors range from light to dark for nitrate and phosphate, whereas it goes from dark to light for silicate, which makes the visual comparison counterintuitive. The same goes for Figure 9.*

We have flipped the silicate colorbar for consistency. Regarding the nitrate colorbar, the reviewer is correct, the inclusion of negative values was due to interpolation artefacts. We

have revised all colorbars.

- *L155: Figure 8 is cited before Figure 7. In this case, the order of these figures should be switched.*

  Yes, Figure 7 and 8 have been switched now.

- *L157-159: It is unclear whether these sentences are referring to Figure 8 or Figure 6.*

  These lines refer to Figure 8 (now Figure 7). We have added figure references for clarification.

  There is evidence of a subtle seasonal cycle in nutrients in the upper 200 dbar of the water column, as shown in Figure 7 (a)–(c). Within each season, there is a large degree of variability. In general, autumn and spring have similar IQR, and winter has the smallest IQR for all nutrients sampled. This suggests winter has the lowest nutrient concentrations available to the surface waters compared to the other seasons. Changes are broadly consistent across the three different nutrients, particularly in the variance and IQR. Although broadly similar, there are differences in the mean nutrient concentrations between the three nutrients with austral season. There is little change to the nitrate mean, although there is an increased number of observations with high concentration in autumn and spring compared to winter (Figure 7 (a)). Phosphate experiences a slight increase in the seasonal mean from autumn through to spring (Figure 7 (b)). Silicate has the largest mean in autumn and winter and the lowest mean in spring (Figure 7 (c)).

- *L167 and L170: Missing the closing brackets for the figure citations.*

  Fixed.

- *L182: "was" should be "is" for consistency with the rest of the text in the section?*

  Fixed.

- *L184-L186: Suggest deleting this paragraph, because it is based on the results not shown and also because part of the results is already mentioned in L175-L177 and the range can be inferred from Figure 7.*

We respectfully disagree with the reviewer on this point. However, we have significantly revised and shortened the paragraph and combined it with the description of Figure 8 for better integration with the relevant results. It now reads:

Similarly to the seasonally grouped sections, EAC jet "inshore" or "offshore" mode sections have the warmest, freshest and most oxygenated water found in the upper ∼50 dbar, and sections cool and have reduced in oxygen concentrations with depth (Figure 8). The TS-O diagram (not shown) reveals little discernible difference between the two modes, as the modes have similar temperature and oxygen ranges.

- *Figure 8: It would be helpful to provide a description for violin plots, as I think not all Ocean Science readers are familiar with violin plots, which seems more complex than others like Taylor diagrams or box plots. At the least, please provide the reference where readers can obtain the necessary information to understand these plots.*

We have switched the figure to a boxplot instead of a violin plot. While we could have expanded on the violin plot, we believe that a simpler boxplot better serves the purpose of conveying the data clearly. We have revised the text substantially to make much better use of the information contained in the new figure. See reply to Reviewer 1 for more detail.

- *L205: "the average properties of the water column" is unclear. Which properties (temperature?) and what does it mean by "average"?*

This has been reworded.

The position of the EAC affects the vertical distribution of nutrients and their longitudinal gradients along the transect.

- *L207: "we see evidence of upwelling" is unclear. Was such evidence shown in Results?*

  This section has been rewritten to clarify our point more effectively. For additional context, see response to major comments and the next comment.

- *L210: "meaning that upwelling in the offshore mode results in higher ..." requires information on the vertical location of the nutrient rich water, rather than the horizontal location mentioned in the previous sentence.*

  We have restructured this part of the discussion for clarity. The discussion now reads:

  While we observe a seasonal nutrient signal, we also highlight the influence of the EAC core's position relative to the continental shelf/slope. The position of the EAC affects the vertical distribution of nutrients and their longitudinal gradients along the transect. For the inshore mode, cooler, fresher, oxygen poor, and nutrient-rich water occurs on the western (inshore) edge of the property sections (west of 154°E), trapped between the EAC and continental slope. Whilst there is a statistical increase in nutrient concentrations, due to the depth (150 dbar) and presence of the EAC jet, these nutrients would be inaccessible to the surface layer. For the offshore mode, there is an uplift of nutrients across the transect, particularly offshore at ∼154.7°E. Additionally, while the MLD is similar between the two modes, there is a shoaling of the nutricline and an increase in nutrients closer to and entering the mixed layer while the EAC is in the offshore mode compared to the inshore mode. The effect of the EAC core location highlights a potential mechanism for episodic nutrient supply to the surface layer.

- *L249: I am not sure if it is ok to bring the not-shown-results into discussion. Is there a reason for not showing the results in the paper? I think it would be beneficial to show such time series comparison even though the lack of temporal coverage. If page limit is an issue, it can be included as supplementary information.*

  We have opted to delete this section entirely, as it diverges too far from the main message of the manuscript.

- *L272: ". M" should be ", m"*

Fixed.

[revised manuscript text omitted]

---

## Author Response (AR2)

**Response to Reviewer Comments on *Jeffers et al*: Control of spatio-temporal variability of ocean nutrients in the East Australian Current**

Megan Jeffers, Christopher Chapman, Bernadette Sloyan, Helen Bostock

**Reviewer Comment:**

- *I thank the authors for thoroughly addressing my earlier comments (Reviewer 2). I am happy to recommend "accept", but just have one little comment on Figure 11 caption:*
  *"A lowess line of fit was also plotted and agreed well with the robust line of fit"*
  *It was not clear to me whether this line was shown on the figure or it was just being checked but not shown. If it is not shown on the figure, I think this sentence can be deleted.*

**Response:**

We thank the reviewer for taking the time to assess the changes we made to our manuscript. The lowess line of fit was plotted to check the different methods for fitting lines of fit to the data, but was not shown in the final figure. The comment "A lowess line of fit was also plotted and agreed well with the robust line of fit" has been deleted from the Figure 11 caption.